



# Secondary organic aerosol formed by EURO 5 gasoline vehicle emissions: chemical composition and gas-to-particle phase partitioning

Evangelia Kostenidou[1,a], Baptiste Marques[1,2], Brice Temime-Roussel[1], Yao Liu[3], Boris Vansevenant[3], Karine Sartelet[4], and Barbara D'Anna[1]

[1]Aix-Marseille Univ, CNRS, LCE, Marseille, France
[2]French Agency for Ecological Transition, ADEME, 49000 Angers, France
[3]AME-EASE, University Gustave Eiffel, Univ Lyon, F-69675 Lyon, France
[4]CEREA, Ecole des Ponts ParisTech, EdF R&D, 77455 Marne-la Vallée, France
[a]Now at: Department of Environmental Engineering, Democritus University of Thrace, 67100 Xanthi, Greece

*Correspondence to*: Evangelia Kostenidou (ekosteni@env.duth.gr) and Barbara D'Anna (barbara.danna@univ-amu.fr)

**Abstract.** In this study we investigated the photo-oxidation of EURO 5 gasoline vehicle emissions during cold urban, hot urban and motorway Artemis cycles. The experiments were conducted in an environmental chamber with average OH concentrations ranging between $6.6 \times 10^5$ - $2.3 \times 10^6$ molecules $cm^{-3}$, relative humidity (RH) 40-55% and temperatures between 22-26ºC. A proton-transfer-reaction time-of-flight mass spectrometer (PTR-ToF-MS) and the chemical analysis of aerosol on-line (CHARON) inlet coupled with a PTR-ToF-MS were used for the gas and particle phase measurements respectively. This is the first time that CHARON inlet was used for the identification of the secondary organic aerosol (SOA) produced from vehicle emissions. The secondary organic gas phase products ranged between $C_1$ and $C_9$ with 1 to 4 atoms of oxygen and were mainly composed of small oxygenated $C_1$-$C_3$ species. The formed SOA contained compounds from $C_1$ to $C_{14}$, having 1 to 6 atoms of oxygen and the products' distribution was centered at $C_5$. Organonitrites and organonitrates contributed 6-7% of the SOA concentration. Relatively high concentrations of ammonium nitrate (35-160 µg $m^{-3}$) were formed. The nitrate fraction related to organic nitrate compounds was 0.12-0.20, while ammonium linked to organic ammonium compounds was estimated only during one experiment reaching a fraction of 0.19. The produced SOA exhibited $logC^*$ values between 2 and 5. Comparing our results to the theoretical estimations, we observed differences of 1-3 orders of magnitude indicating that additional parameters such as RH, particulate water content, aerosol hygroscopicity, and possible reactions in the particulate phase may affect the gas-to-particle partitioning.

## 1 Introduction

SOA is formed in the atmosphere through chemical reactions and constitutes a major part of the OA (Jimenez et al., 2009; Pandis et al., 2016). Even though biogenic SOA often dominates over anthropogenic SOA at a global scale (Kanakidou 2005), SOA formed from the photo-oxidation of aromatic hydrocarbons is estimated at 33% (Kelly et al., 2018). In highly




urbanized and populated areas anthropogenic SOA may dominate over the biogenic SOA (e.g., Volkamer et al., 2006; Platt et al., 2014) as aromatic hydrocarbons have been considered among the major precursors of SOA in urban environments (Hayes et al., 2015; Wu and Xie, 2018).

Emissions from motor vehicles are an important source of urban air pollution, emitting both particulate and gas phase
species (Dallmann and Harley, 2010, Borbon et al., 2013; Platt et al., 2014; Argyropoulos et al., 2016; Hofman et al., 2016; Gentner et al., 2017; Kostenidou et al., 2021). PM vehicle emissions mainly contain primary organic compounds (POA) of low and semi-volatile species as well as black carbon (BC) while gas phase vehicle emissions include $CO_2$, CO, $NO_x$, $NH_3$ and volatile organic compounds (VOCs) (Gordon et al., 2013; Platt et al., 2103; Saliba et al., 2017; Pieber et al., 2018). Concerning gasoline vehicles, among the most important emitted VOCs that serve as SOA precursors are $C_6$ - $C_{11}$ light
aromatics, naphthalene, methyl-naphthalenes (Nordin et al., 2013; Liu et al., 2015; Saliba et al., 2017; Pieber et al., 2018). Intermediate volatile compounds (IVOCs) may also be emitted by both diesel and gasoline motor vehicles (Zhao et al., 2015; Drozd et al., 2019; Marques et al., 2022). Some VOCs and IVOCs are of significant relevance since they are efficient particle precursors upon reaction with atmospheric oxidants (such as OH radicals, $NO_3$ radicals and $O_3$) forming SOA and thus increasing the particulate matter (PM) budget.

Gasoline vehicles may dominate over diesel vehicles in the SOA production in urban areas (Bahreini et al., 2012; Jathar et al., 2017). Platt et al. (2017) showed that both GDI and PFI gasoline vehicles produce significantly higher SOA, compared to the latest generation diesel vehicles equipped with a diesel oxidation catalyst (DOC) and a diesel particulate filter (DPF). The high SOA formation from gasoline vehicles has also been confirmed in several studies (e.g., Platt et al., 2013; Nordin et al., 2013; Gordon et al., 2014; Liu et al., 2015; Karjalainen et al., 2016; Saliba et al., 2017; Zhao et al., 2017; Ma et al., 2018;
Pieber et al., 2018; Simonen et al., 2019; Roth et al., 2019; Hartikainen et al., 2023). The produced SOA from gasoline vehicles can be 1-2 orders of magnitude higher than the emitted POA (Platt et al., 2013; Gordon et al., 2014; Liu et al., 2015; Suarez-Bertoa et al., 2015; Karjalainen et al., 2016; Pieber et al., 2018; Kari et al., 2019).

The chemical composition of the formed SOA highly depends on the gas phase emissions. A large fraction of gasoline vehicles emissions is composed of xylenes, ethylbenzene, trimethylbenzenes, ethyltoluene, toluene, benzene and naphthalene
(e.g., Zimmerman et al., 2016; Saliba et al., 2017; Marques et al., 2022). Thus, it is expected that SOA produced by gasoline vehicles emissions is dominated by the SOA products of the above precursors. Some of the most common SOA products of the those compounds are methylglyoxal, glyoxal, maleic anhydride, 4-oxo-pentenal, 4-oxo-butenoic acid, succinic anhydride, citraconic anhydride, 4-(hydroxymethyl)-2-furaldehyde, acetic acid, glyoxylic acid, 4-oxo-butenoic acid, malonic acid (Forstner et al., 1997; Smith et al., 1998; 1999; Jang and Kamens 2001; Cocker Iii et al., 2001; Hamilton et al., 2003;
2005; Zhao et al., 2005; Huang et al., 2006; Wang et al., 2006; Sato et al., 2007; 2012; Healy et al., 2008; Wyche et al., 2009; Borrás and Tortajada-Genaro, 2012; White et al., 2014; Wu et al., 2014; Ma et al., 2018; Schwantes et al., 2017). The above studies used analytical techniques such as gas chromatography mass spectrometry (GC-MS) and liquid chromatography mass spectrometry (LC-MS) for compounds identification. Even though, there are many studies describing





the SOA chemical composition by individual precursors found in the gasoline vehicles exhaust, very little is known about

the chemical composition of the bulk SOA produced by gasoline vehicles exhausts (Platt et al., 2013; Nordin et al., 2013; Pieber et al., 2018; Hartikainen et al., 2023).

Except for the chemical composition, the partitioning between the gas and particle phase is a key parameter as it defines weather a compound resides in the particle or in the gas phase and thus its lifetime in the atmosphere and its reactivity. The most common way to derive the gas-to-particle phase partitioning at a molecular level is the extraction of filters and

denuders with particle and gas phase samples and the corresponding analysis in analytical devices such as GC-MS, LC-MS, etc. For example, this method has been applied to SOA from photo-oxidation of toluene (Jang and Kamens 2001), 1,3,5-trimethylbenzene (Healy et al., 2008; Praplan et al., 2014) and m-xylene (Leach et al., 1999). During the last decade the volatility of the bulk SOA has been extensively characterized by heated laminar flow reactors, (e.g., thermodenuders) in terms of thermograms. The anthropogenic SOA volatility has been described in various studies (e.g., Hildebrandt et al.,

2009; 2015; Huffmann et al., 2009; Kim et al., 2013; Kolesar et al., 2015; Docherty et al., 2018; Li et al., 2018; Sato et al., 2019). Other groups have estimated the volatility distribution as a function of the saturation concentration $C^*$ either combining experimental and model calculations (Robinson et al., 2007) or using directly experimental data (Hinks et al., 2018; Sato et al., 2019). The above approaches have been extensively applied to laboratory biogenic SOA systems (e.g., Saha and Grieshop, 2016; Kostenidou et al., 2018a), to anthropogenic OA sources such as cooking OA (COA) (Louvaris et

al., 2017a), to ambient sources derived by source apportionment algorithms (e.g., Cappa and Jimenez, 2010; Paciga et al., 2016; Louvaris et al., 2017b; Kostenidou et al., 2018b) but they are limited to aromatic SOA.

The last decade, online high-resolution instrumentation has been developed for the identification of the OA species at a molecular level (Williams et al., 2006, 2014; Kreisberg et al., 2009; Hohaus et al., 2010; Zhang et al., 2014) or for the identification of the chemical formula using soft-ionization mass spectrometry (Lopez-Hilfiker et al., 2014; Isaacman-

VanWertz et al., 2017; Stark et al., 2017; Gkatzelis et al., 2018; Lannuque et al., 2023). Volatility measurements are performed by alternative measurements between the gas and the particle phase of the same ion (Hohaus et al., 2015; Lopez-Hilfiker et al., 2016; Isaacman-VanWertz et al., 2016; Stark et al., 2017; Gkatzelis et al., 2018; Lannuque et al., 2023). To our knowledge these new techniques are still limited. They have been applied to biogenic OA systems in the ambient (Lopez-Hilfiker et al., 2016; Isaacman-VanWertz et al., 2016; Stark et al., 2017) or in laboratory produced biogenic SOA

(Hohaus et al., 2015; Gkatzelis et al., 2018), and just recently on toluene SOA (Lannuque et al., 2023).

In this work the gas and the particle phase of the secondary organic species produced by photo-oxidation of EURO 5 gasoline vehicle emissions were studied. The photo-oxidation took place in an environmental chamber irradiated for several hours. We used real time high-resolution instrumentation such as proton-transfer-reaction time-of-flight mass spectrometer (PTR-ToF-MS) combined with a chemical analysis of aerosol on-line (CHARON) inlet and a high-resolution time-of-flight

aerosol mass spectrometer (HR-ToF-AMS). We investigated the fresh and the oxidized gas phase chemical composition as well as the SOA chemical composition based on their chemical formula. In addition, we calculated the saturation



concentration of the major SOA species as a function of their chemical formula, and we provided the saturation concentration of specific compounds identified in the SOA. This is the first time that the CHARON/PTR-ToF-MS system was applied to SOA formed by gasoline vehicles emissions, and this is the first study that provides volatility information based on chemical formula for gasoline emissions SOA.

## 2 Experimental set up and instrumentation

A gasoline direct injection (GDI) Euro 5 light duty vehicle of 1.2 TCE 16V size class, equipped with a three-way catalyst (TWC), with an engine capacity of 1149 $cm^3$ and mileage of 97089 Km was used. The vehicle was rented from a local rental car office and fueled by standard unleaded 95-E10 gasoline. The experiments were conducted in the facilities of the Environment, Planning, Safety and Eco-design Laboratory (EASE) of the Gustave Eiffel University. The vehicle was tested running Artemis cold urban, hot urban and motorway cycles on a roll-bench chassis dynamometer; detailed characterization of the fresh gas and particle phase emissions of the same vehicle are described in Kostenidou et al. (2021) and Marques et al. (2022). Fresh emissions were transferred to an 8 $m^3$ custom-made Teflon chamber (Louis, 2018) through a 2m stainless steel heated (at 120 ºC) line using a dilution ejector of one stage with a dilution ratio (DR) of 1.5. Five chamber experiments were conducted in total (Table 1). For experiments 1 and 3 only the first 5 minutes of the cycle were injected inside the chamber, while for the remaining experiments the whole cycle was introduced.

Comparing the concentrations of the major ions detected by PTR-ToF-MS inside the chamber (Table 2) 10-15 min after the filling of the chamber and those at the exhaust of the vehicle (Marques et al., 2022) we calculated a dilution ratio ranging from 28 to 112 (Table 1). After filling the chamber with fresh emissions, an extra dilution of 2.3-5.1 was applied in 3 out of 5 experiments before the oxidation procedure (Table 1). Thus, the total dilution ratio in the chamber was 48-518 (Table 1).

The fresh emissions were stabilized for 1-2 hours inside the chamber; then OH radicals were produced by $H_2O_2$ photolysis under UV illumination at a 280-320 nm wavelength. $H_2O_2$ solution (50 wt. % in $H_2O$, Sigma-Aldrich) was introduced into the chamber through a bubbler. The OH concentration was calculated by the decays of 1,3,5-trimethylbenzene (TMB) and m-xylene using the rate constants of $5.7x10^{-11}$ and $2.3x10^{-11}$ $cm^3$ molecule$^{-1}$ s$^{-1}$ respectively (Calvert et al., 2002). An average OH radical concentration was evaluated considering the two decays and depending on the experiment it ranged from 6.6 $x10^5$ to $2.3x10^6$ molecules cm$^{-3}$ (Table 1). The total OH exposure ranged between $6.2x10^9$ and $1.6x10^{10}$ molecules cm$^{-3}$ s (Table 1). During the experiments, the relative humidity (RH) inside the chamber was 45-55%, while the temperature was 22-26ºC.

The gas phase was measured by a PTR-ToF-MS (PTR-ToF 8000 instrument Ionikon Analytik) (de Gouw and Warneke, 2007) with a time resolution of 30s. The PTR-ToF-MS was operated at an electrical field (E/N) of 100 Td, the drift tube pressure was 2.2 mbar, while the inlet tube and the reaction chamber were at 120˚C. PTR-ToF-MS was calibrated using a multicomponent gas standard (Apel Riemer Environmental Inc., Miami, FL, USA) which includes low molecular weight

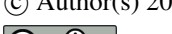



analytes (*m/z'* s<200). The organic particle phase was measured by a CHARON inlet (Eichler et al., 2015) coupled in front of the PTR-ToF-MS (this system will be referred to as "CHARON" hereafter. The recently developed CHARON inlet
consists of a gas-phase denuder for stripping off the gas phase molecules. The particles then pass through a series of aerodynamic lenses for particle collimation and pre-concentration and through a thermo-desorption unit, where they are volatilized and transferred to the PTR-ToF-MS detector. The CHARON inlet was operated at low pressure (6-8 mbar) and the thermo-desorption unit at a constant temperature of 150°C, while the E/N and drift tube pressure were maintained at 100 Td and 2.2 mbar correspondingly. The sample time resolution was 30s. During a single experiment of photo-oxidation we
switched between the CHARON inlet and the gas phase mode 3-4 times. Particles were sampled at least for 40 min and only the last 5 min of each sample were considered as the stabilization required at least 20-25 min.

For the particle chemical characterization, we additionally deployed an Aerodyne HR-ToF-AMS (DeCarlo et al., 2006; Canagaratna et al., 2007).

## 3 Methods

The PTR-ToF-MS data were analyzed using the Tofware v2 for peak fitting and the PeTeR v3.5 toolkit for cps to ppb conversion with Igor Pro 6.37 (Wavemetrics). For the assignment of the secondary VOCs and SOA compounds we used as a guide the molecular formula of the products that have been identified in previous studies from aromatic SOA photo-oxidation systems (Forstner et al., 1997; Smith et al., 1998; 1999; Jang and Kamens 2001; Cocker Iii et al., 2001; Hamilton et al., 2003; 2005; Zhao et al., 2005; Huang et al., 2006; Wang et al., 2006; Sato et al., 2007; 2012; Healy et al., 2008;
Wyche et al., 2009; Müller et al., 2012; Borrás and Tortajada-Genaro, 2012; Wu et al., 2014; White et al., 2014; Wu et al., 2014; Ma et al., 2018). For the raw counts to ppb conversion, we used the experimentally determined transmission function and k proton rate constants based on the molecular formula of each ion. These k rates ranged from $1.4\times10^{-9}$ to $3.9\times10^{-9}$ cm$^3$ s$^{-1}$ and were close to previous literature (e.g., Anicich 2003; Warneke et al., 2003; Cappellin et al., 2012; Holzinger et al., 2019). More details about the quantification of the PTR-ToF-MS signal are discussed in the SI.
During our experiments high ammonium nitrate levels were produced resulting in nitrate concentrations up to 131 µg m$^{-3}$. Thus, the AMS data were corrected modifying the fragmentation table according to the suggestions of Pieber et al. (2016). As detailed in the above paper, high concentrations of nitrate salts interfere to the measurement of particulate $CO_2^+$, by creating an oxidizing environment on the particle vaporizer in the HR-ToF-AMS and thereby releasing $CO_2^+$ from carbonaceous residues. This affects the calculated OA mass concentration, the mass spectra and the O:C ratio.
The experimental saturation mass concentration ($C_i^*$) for each of the identified compounds was calculated based on the gas-to-particle phase partitioning theory (Pankow, 1994) as described by Donahue et al. (2006) Eq. (1):

$$C_i^* = TS\frac{C_{g,i}}{C_{p,i}} \tag{1}$$



where, $C_{p,i}$ and $C_{g,i}$ are the particle and gas phase mass concentrations correspondingly (in µg m$^{-3}$) of the species i and TS is the mass concentration of the total suspended organic and inorganic aerosol (in µg m$^{-3}$). The gas phase concentration ($C_{g,i}$) of

each compound was calculated from PTR-ToF-MS data, while the corresponding particle phase ($C_{p,i}$) concentration was measured by CHARON, after normalizing the total OA CHARON mass concentration to the total HR-ToF-AMS OA mass concentration. The OA mass concentrations of HR-ToF-AMS and CHARON may differ between each other up to a factor of 2 as mentioned in Müller et al. (2017) due to fragmentation of analyte ions in PTR-ToF-MS.

The total suspended mass concentration was derived by the HR-ToF-AMS (OA+ammonium nitrate). Due to the high

ammonium nitrate mass concentration and the increased RH (~50%) the HR-ToF-AMS collection efficiency (CE) was assumed close to unity (Matthew et al., 2008).

The saturation mass concentration, $C_i^*$ (Donahue et al., 2011) is linked to $K_{p,i}$ through the Eq. (2):

$$C_i^* = \frac{1}{K_{p,i}} \tag{2}$$

The theoretical saturation concentration was estimated using Eq. (3) proposed by Cappa and Jimenez (2010):

$$C_i^*(T) = \frac{MW_{OA,i} \times 10^6 \times p_{i,L} \times \zeta_i}{R \times T} \tag{3}$$

where, $MW_{OA,i}$ is the molecular weight of the organic compound i, (g mol$^{-1}$), $p_{i,L}$ is the subcooled liquid saturation vapor pressure (Pa), $\zeta_i$ is the activity coefficient of the compound i in the particulate phase, T is the temperature inside the chamber (K) and R is the ideal gas constant (8.314 J mol$^{-1}$ K$^{-1}$). The activity coefficient was assumed to be equal to unity. The molecular weight of each compound was determined by the *m/z* detected by CHARON or PTR-ToF-MS (M+1)$^+$ as the

parent compounds gain a proton. The subcooled liquid saturation vapor pressure ($p_{i,L}$) was estimated by empirical relationships based on the equation of Clausius-Clapeyron (Myrdal and Yalkowsky, 1997; Jenkin, 2004; Nannoolal et al., 2008). The above approaches require the boiling temperature and the enthalpy of vaporization; these properties were estimated based on the molecular structure of each compound (Mackay et al., 1982; Joback and Reid, 1987; Stein and Brown, 1994). For the prediction of the vapor pressure of each compound we used the online facility UmaSysProp

(https://umansysprop.readthedocs.io/en/latest/) developed by Topping et al. (2016) using as input the molecular information in terms of SMILES strings (Simplified Molecular Input Line Entry System). On this facility, there are three options for the boiling temperature estimation based on the methods of Joback and Reid (1987), Stein and Brown (1994) and Nannoolal et al. (2008). After having selected the method for the boiling temperature estimation the user has the choice to select two approaches for the vapor pressure prediction (Myrdal and Yalkowsky, 1997; Nannoolal et al., 2008). Finally, the above tool

also includes the EVAPORATION method (Compernolle et al., 2011). Therefore, there are seven different combinations for the vapor pressure estimation. For the comparisons with our measurements, we used the average vapor pressure of these seven combinations for each compound.

For the estimation of the organic nitrate fraction, $x_{organonitrate}$, (i.e., the organic nitrate mass concentration over the total nitrate mass concentration) we applied the procedure of Farmer et al. (2010) on the AMS data:



$$x_{organonitrate} = \frac{ONit}{TotNit} = \frac{(1+R_{ON}) \times (R_{[NO^+/NO_2^+]meas} - R_{[NO^+/NO_2^+]NH_4NO_3cal})}{(1+R_{NO^+/NO_2^+]meas}) \times (R_{ON} - R_{[NO^+/NO_2^+]NH_4NO_3cal})}$$ (4)

where $R_{[NO^+/NO_2^+]NH_4NO_3cal}$ is the ratio of $NO^+/NO_2^+$ ions during $NH_4NO_3$ calibrations (0.63 on average), $R_{[NO^+/NO_2^+]meas}$ is the measured ratio of $NO^+/NO_2^+$ ions throughout the experiment and $R_{ON}$ is a fix value set to 22.9, equal to the maximum ratio of $NO^+/NO_2^+$ observed.

## 4 Results

**4.1 Gas phase**

In total 59-67 (for the cold/hot urban cycles) and 103 (for the motorway cycle) ions were detected during the characterization of the fresh VOCs after their introduction in the chamber (Table S1). 47-75 of those ions had a contribution higher than 0.14% to the total fresh gas phase concentration, which explained 96-99% of the measured concentration (Table S1). Table S2 lists the major ions of those having a contribution greater that 0.14% to the total fresh gas phase concentration. Any ions
detected at $m/z$ 39 (e.g., $C_3H_3^+$) were excluded from our analysis as $m/z$ 39 was dominated by the water cluster isotope $H_5O[18O]^+$ which could possibly lead to artifactually high concentrations of the rest ions detected at $m/z$ 39.

The O and C carbon distributions of the fresh VOCs for the 3 different cycles as measured by the PTR-ToF-MS are illustrated in Figures 1a, 2a and 3a. Previous studies (e.g., Saliba et al., 2017) have shown that small alkanes < $C_5$ and alkenes < $C_2$, may contribute up to 50% of the non-methane VOC (NMVOC) emissions for gasoline vehicles. As PTR-ToF-
MS cannot detect these small compounds, due to their low proton affinity, the fresh VOCs emissions were mainly composed of $C_6$-$C_{10}$ aromatic compounds: 52-58% for the cold and hot urban cycles and 38% for the motorway cycle (Table 2). For the cold urban cycles, $C_8$ aromatics (e.g., xylenes, ethylbenzene) were the major aromatic compounds (24-26%) followed by $C_9$ aromatics (e.g., trimethylbenzenes, 1-ethyl-3-methylbenzene) (14.0%), toluene (9.6%) and benzene (4.8%). The hot urban cycle was characterized by lower contribution of $C_8$ and $C_9$ aromatics (16.4% and 8.4% respectively) but higher fraction of
benzene (13.2%) and toluene (11.7%) in comparison to the cold urban cycles. During the motorway cycle $C_9$, and $C_8$ aromatic compounds were the major contributors (10.9% and 12.3% respectively), followed by benzene (7.1%), toluene (4.6%) and $C_{10}$ aromatics (3.4%). Fragments of large alkane compounds or alkenes ($C_xH_y^+$) were detected at lower $m/z$'s (between 41 and 71) with fractions ranging between 20-26% for all the cycles. Small, oxygenated compounds such as formaldehyde, methanol, acetaldehyde, ethanol, acrolein and acetone) were also emitted mainly during the cold urban cycles.
All the above compounds have been identified in gasoline exhaust emissions (Erickson et al., 2014; Gueneron et al., 2015; Pieber et al., 2018; Kari et al., 2019).

A detailed analysis of the primary VOC emissions and assignment from this vehicle are discussed by Marques et al. (2022). Briefly fresh online emissions of this vehicle as seen by the PTR-ToF-MS and GC-MS were characterized by monoaromatic compounds (62%) centered around $C_9$. The above fraction was slightly higher compared to that measured inside the chamber




(this study), where the monoaromatic compounds were on average 52%. This difference could be due to potential losses on the tubing and chamber walls. In addition, Marques et al. (2022) reported that aliphatic compounds were also an important fraction (35%). Linear and branched alkanes were predominant (79%), but alkenes represented a non-negligible part (17%) of the aliphatic compounds.

Secondary VOCs accounted for 92 – 163 identified ions (Table S1) (at OH exposure $5.2 \times 10^9$-$1.6 \times 10^{10}$ molecules cm $^{-3}$ s,

Table 1). Approximately two thirds of these ions had a contribution higher than 0.14% to the total secondary gas phase concentration, explaining 93-97% of the secondary gas phase concentration (Table S1). Table S3 presents a sub-group of the ions with a contribution greater that 0.14% to the total secondary gas phase concentration, which explained 89-97% of total secondary gas phase concentration.

The O and C carbon distributions of the secondary volatile compounds produced for the 3 different cycles are showed in

Figures 1b, 2b and 3b. For the calculation of these distributions, we subtracted the amounts of the $C_1$, $C_2$ and $C_3$ oxygenated compounds found in the primary emissions and the non-oxygenated compounds/fragments ($C_6$-$C_{10}$ aromatics compounds and $C_xH_y^+$ fragments) that did not react. The major products, which explain 83-94% of the produced secondary VOCs concentration and their possible assignment are presented in Table 3.

A large fraction was composed of small oxygenated species with 1 to 2 atoms of oxygen distributed between $C_1$ and $C_3$.

These compounds were mainly products of small alkanes and alkenes (e.g., $CH_4$, $C_2H_4$, $C_2H_6$) which can be as high as 50% of the total VOC emissions, but they are not measured by the PTR-ToF-MS. These compounds react with OH radicals generating high amounts of $C_1$-$C_3$ products. However, part of the signal detected at these *m/z's* could be also fragments from larger molecules. The *m/z's* 31.02 $(CH_2O)H^+$, and 47.01 $(CH_2O_2)H^+$ possibly formaldehyde and formic acid or fragments from larger compounds, were the major species with one atom of carbon. The main compounds with 2 atoms of carbons

were *m/z's* 45.03 $(C_2H_4O)H^+$ acetaldehyde, 61.03 $(C_2H_4O_2)H^+$, acetic acid/ hydroxyacetaldehyde (together with their fragments at *m/z* 43.02 $C_2H_3O^+$) and 77.02 $(C_2H_4O_3)H^+$ glycolic acid/PAN fragment. However, part of the smaller *m/z's* such as *m/z's* 45 and 61 are often fragments of larger compounds (e.g., esters) (Burh et al., 2002; Haase et al., 2012; Baasandorj et al., 2015). The *m/z* 59.05, $(C_3H_6O)H^+$, which could be acetone or a fragment (Buhr et al., 2002; Karl et al., 2007) is the species with the highest contribution among the $C_3$ compounds. It is followed by *m/z* 73.03 $(C_3H_4O_2)H^+$ methylglyoxal

(together with its fragment at *m/z* 45.03 $(C_2H_4O)H^+$ and its C13 isotope at *m/z* 74.03), *m/z* 57.03, $(C_3H_4O)H^+$ acrolein and *m/z* 75.04, $(C_3H_6O_2)H^+$ hydroxyacetone/ propanoic acid/fragment.

Other important contributions were due to *m/z's* 71.05 $(C_4H_6O)H^+$ 2-butenal, 73.06, $(C_4H_8O)H^+$ butanone, 87.04 $(C_4H_6O_2)H^+$ 3-oxobutanal and isomers, 107.05 $(C_7H_6O)H^+$ benzaldehyde, 113.02 $(C_5H_4O_3)H^+$ citraconic anhydride (methylfurandione), 113.06 $(C_6H_8O_2)H^+$ 2-methyl-4-oxo-pentenal and more, 121.07 $(C_8H_8O)H^+$ acetophenone/m-toluraldehyde, 99.04

$(C_5H_6O_2)H^+$ angelicatone/4-oxo-2-pentenal (isomers)/2-methylbutenedial/1,3-cyclopentanedione/2(3H)-furanone, 5-methyl-(isomers), 127.04 $(C_6H_6O_3)H^+$ hydroxymethyl furaldehyde/ dimethylmaleic anhydride/ hydroxy-methyl-pyranone, 129.06 $(C_6H_8O_3)H^+$ methyl–4–oxo–2–pentenoic acid/ hydroxy-oxo-hexenal/ furanone, 141.06, $(C_7H_8O_3)H^+$ oxo-heptedienoic



acid/epoxy-methyl-hexenedial. It must be mentioned that Tables 3 and S3 do not take into account any fragmentation in the concentration calculations unless it is mentioned.

Most of the above *m/z's* have been detected in the photo-oxidation gas-phase products of 1,3,5-TMB (Healy et al., 2008; Müller et al. 2012) and toluene (Jang and Kamens 2001; Schwantes et al., 2017, Lannuque et al. 2023). It should be mentioned that most of the ions reported by Lannuque et al. (2023), which studied the toluene photo-oxidation SOA using CHARON, were detected in the present work. These findings suggest that the monoaromatics precursors emitted from the vehicle exhaust play a predominant role in the secondary VOC formation.

The secondary VOCs produced by the three different cycle emissions presented similar O and C distributions between each other (Figures 1b, 2b and 3b). However, differences were observed between the cycles especially for the smaller ions. The secondary gas phase products from cold cycles emissions had a higher fraction of $C_2$ and $C_3$ compounds with 1 atom of oxygen but lower fraction of $C_2$ compounds with 2 atoms of oxygen compared to the secondary gas phase produced by hot urban and motorway cycles emissions. These differences could be attributed to the different primary gas phase emissions as

mentioned in a previous section. For the vehicle of this study Marques et al. (2022) also reported lower fraction of $C_8$ branched alkanes during the cold urban cycle. Those variations in emission composition between the different type of cycles slightly affect the secondary VOCs O and C distributions.

## 4.2 Particle phase

Primary organic aerosol (POA) had an aerodynamic mode diameter of around 100 nm (as measured by the AMS) (Figure 4a)

and therefore it could not be detected by CHARON. During the photo-oxidation of the gasoline emissions, along with the SOA a relatively high mass concentration of ammonium nitrate was produced with fractions to the total particle phase 0.74-0.93 (Table S4). This is confirmed by previous gasoline vehicle SOA studies (e.g., Roth et al., 2019; Simonen et al., 2019) as elevated $NH_3$ and $NO_x$ emissions are released from gasoline vehicle engines. Ammonium, nitrate and SOA had the same size distribution with aerodynamic mode diameters around 450 nm (Figure 4b) so that they could be efficiently detected by

CHARON.

The CHARON sampling periods were taken at OH exposures in the range $5.2 \times 10^9$-$1.6 \times 10^{10}$ molecules cm$^{-3}$ s (Table 1). Depending on the experiment, 169 to 253 ions were detected in the particulate phase (Table S1). The ions with a contribution higher than 0.14% to the total SOA explained 92-95% of the SOA (Table S1). Table S5 lists a sub-group of the detected ions corresponding to a SOA contribution higher than 0.14% representing 79-86% of the overall measured particle phase

concentration. Even though SOA included ion fragments with *m/z's* up to 300, the fraction between *m/z's* 200 and 300 was very low (below 2%) (Table S4). Table 4 provides the most significant SOA ion fragments and their tentative assignment based on literature work (Forstner et al., 1997; Smith et al., 1998; 1999; Jang and Kamens 2001; Cocker Iii et al., 2001; Hamilton et al., 2003; 2005; Zhao et al., 2005; Huang et al., 2006; Wang et al., 2006; Sato et al., 2007; 2012; Healy et al., 2008; Wyche et al., 2009; Müller et al., 2012; Borrás and Tortajada-Genaro, 2012; Wu et al., 2014; White et al., 2014; Wu et



al., 2014; Ma et al., 2018). The ion fragments in Table 4 represent 35-65% of the SOA concentration fraction. Tables 4 and S5 do not account for any fragmentation in the concentration calculations unless it is mentioned.

The O and C distributions of the SOA are shown in Figures 1c, 2c and 3c. The SOA products were distributed between $C_1$ and $C_{14}$ having up to 6 atoms of oxygen. $m/z$ 47.01 $(CH_2O_2)H^+$ formic acid along with $m/z's$ 31.02 $(CH_2O)H^+$ formaldehyde and possibly fragments are practically the whole fraction of the $C_1$ species. The $C_2$ products contained $m/z$ 43.02 $C_2H_3O^+$ and

$m/z$ 45.03 $(C_2H_4O)H^+$, two common fragments of larger compounds, ion fragment and $m/z$ 61.03 $(C_2H_4O_2)H^+$ hydroxy acetaldehyde/acetic acid ion fragment. $m/z$ 73.03 $(C_3H_4O_2)H^+$ was tentatively assigned to methylglyoxal and had an appreciable contribution of 2.9-4.6% of the formed SOA (together with its fragment at $m/z$ 45.03 $(C_2H_4O)H^+$ and its C13 isotope at $m/z$ 74.03). Other important $m/z's$ were 85.03 $(C_4H_4O_2)H^+$ butenedial (and isomers)/furan-2-one (and isomers), 87.04 $(C_4H_6O_2)H^+$ e.g., 3-oxobutanal, 89.02 $(C_3H_4O_3)H^+$ e.g., methyl glyoxylic acid, 97.03 $(C_5H_4O_2)H^+$ e.g., furfural, 99.01

$(C_4H_2O_3)H^+$ e.g., maleic anhydride, 99.04 $(C_5H_6O_2)H^+$ e.g., 4-oxo-2-pentenal (and isomers), 101.02 $(C_4H_4O_3)H^+$ e.g., succinic anhydride, 103.04 $(C_4H_6O_3)H^+$ e.g., hydroxy-oxobutanal (and isomers), 111.04 $(C_6H_6O_2)H^+$ e.g., methylfuraldehyde (and isomers), 113.02 $(C_5H_4O_3)H^+$ e.g. methylfurandione, 115.04 $(C_5H_6O_3)H+$ e.g., 4-oxo-2-pentenoic acid (and isomers), 117.02 $(C_4H_4O_4)H^+$ e.g., maleic acid, 127.04 $(C_6H_6O_3)H+$ e.g., 4-(hydroxymethyl)-2-furaldehyde, 129.06 $(C_6H_8O_3)H^+$ e.g., methyl–4–oxo–2–pentenoic acid, 141.06 $(C_7H_8O_3)H^+$ e.g., oxo-heptedienoic acid/epoxy-methyl-hexenedial, 153.06

$(C_8H_8O_3)H^+$ e.g., hydroxy-dimethyl-cyclohexadiene-dione, 155.07 e.g., $(C_8H_{10}O_3)H^+$ e.g., methyl-heptene-trione (Forstner et al., 1997; Smith et al., 1998; 1999; Jang and Kamens 2001; Cocker Iii et al., 2001; Hamilton et al., 2003; 2005; Zhao et al., 2005; Huang et al., 2006; Wang et al., 2006; Sato et al., 2007; 2012; Healy et al., 2008; Wyche et al., 2009; Müller et al., 2012; Borrás and Tortajada-Genaro, 2012; Wu et al., 2014; White et al., 2014; Wu et al., 2014; Ma et al., 2018). Most of detected ions identified in the particle phase in this work, have been reported in the study of Lannuque et al. (2023), where

CHARON was used for the identification of the toluene photo-oxidation SOA.

Generally, the SOA formed by the three different driving cycles had similar O and C distributions (Figures 1c, 2c and 3c) with some minor differences, however. For example, SOA from cold and hot urban cycles emissions had a higher fraction of $C_1$ and $C_2$ compounds with 1-3 atoms of oxygen in comparison to the SOA derived by motorway cycles emissions. Similar to the secondary gas phase products, the differences in the SOA distributions are probably due to the differences in the

primary gas phase emissions among the various cycles.

Organonitrates and organonitrites (ON) were identified in the SOA products. Their contribution was 6-7% of the total SOA concentration (Table S4). The most important ON ion fragments were detected at the $m/z's$ 60.04 $(C_2H_5NO)H^+$, 74.02 $(C_2H_4NO_2)H^+$, 90.02 $(C_2H_3NO_3)H^+$, 112.04 $(C_5H_5NO_2)H^+$, 138.05 $(C_7H_7NO_2)H^+$ (nitrotoluene) 140.03 $(C_6H_5NO_3)H^+$ (nitrophenol), 154.05 $(C_7H_7NO_3)H^+$ (nitrocresol), 156.03 $(C_6H_5NO_4)H^+$ (nitrocatechol), 168.07 $(C_8H_9NO_3)H^+$ (ethyl-

nitrophenol/ dimethyl-nitrophenol) and 170.05 $(C_7H_7NO_4)H^+$ (methylnitrocatechol) (Tables 4, Table S5 and Figure S1). Some of the above ON compounds (e.g., nitrotoluene, nitrophenol, nitrocresol) have been detected in the particulate phase during the photo-oxidation of benzene, toluene, benzene, xylene (e.g., Forstner et al, 1997; Jang and Kamens, 2001;



Hamilton et al., 2005; Borras et al., 2012; Sato et al., 2012; Schwantes et al., 2017; Lannuque et al., 2023). The majority of the nitrogen organic compounds contained one atom of N and ranged between $C_1$ and $C_{14}$ for the experiment 1 (cold urban)

and between $C_1$-$C_{12}$ for the other cycles (Figure S2). They contained 1-5 atoms of oxygen (Figure S3). This non-negligible amount of nitrogen organic compounds is probably due to the combination of the elevated RH and high $NO_x$ concentrations (90-2000 ppb) in our experiments. Lim et al. (2016) found that a high fraction of oxygenated organonitrates can be formed in the presence of $NO_x$ and high RH (i.e., wet aerosols) during glyoxal photo-oxidation. In another study Jiang et al. (2019) found that the organic nitrate compounds signal in the SOA increased with the RH during furan photo-oxidation.

Using equation (4) and the HR-ToF-AMS data we calculated the organic nitrate fraction to the total nitrate mass concentration for the five experiments for the period that the CHARON sample was taken. This fraction ranged from 0.12 to 0.20 of the total nitrate concentration (Table S4). For the experiment #5 (photo-oxidation of motorway emissions) we applied equation (4) throughout the whole experiment since there were no clogging issues at the HR-ToF-AMS inlet. The time series of inorganic and organic nitrate concentration are depicted in Figure S4, while the organic nitrate fraction is

shown in Figure S5. The organic nitrate fraction remains stable during the experiment with an average value of 0.19±0.02.

We investigated the ratio cations/anions (Table S4) of inorganics taking into account the organic nitrate contribution for the different experiments for the period that the CHARON sample was taken. For experiments #2 and 3 (both cold urban emissions) the particles were practically neutralized. For the experiments # 1 and 4, the inorganic particles were acid indicating a deficit of $NH_4^+$ or excess of $NO_3^-$. This could be explained by the uptake of $HNO_3$ in the particulate phase due to

the high RH, which turned the particles into rather liquid phase. These two experiments indeed had the highest nitrate concentrations (110 and 131 µg m$^{-3}$). Melhman et al. (1995) noticed a deficit of cations in ambient measurements, when the nitrate levels were relatively high (higher than 11 µg m$^{-3}$). They attributed this behavior to $HNO_3$ absorption into the particulate phase which had a high liquid water content (RH>80%).

In the experiment #5 the particles showed an excess of $NH_4^+$. The ammonium fraction that is not neutralized by inorganic

anions could be explained by the presence of organic salts of ammonium using the equation below:

$$x_{organoammonium} = \frac{[NH_4^+]_{meas} - (1 - x_{organonitrate}) \times [NO_3^-]_{meas} \frac{18}{62}}{[NH_4^+]_{meas}} \qquad (5)$$

where $x_{organoammonium}$ is the organic ammonium fraction (i.e., the organic ammonium mass concentration over the total ammonium mass concentration), $[NH_4^+]_{meas}$ and $[NO_3^-]_{meas}$ is the ammonium and nitrate mass concentrations respectively as measured by the AMS. Applying equation (5) to experiment #5 (photo-oxidation of motorway emissions) for the period

that the CHARON sample was taken, we found that a fraction of 13% of the ammonium was in the form of organic ammonium salt (Table S4). Using equation (5) throughout the whole experiment #5 we estimated the organic and inorganic ammonium mass concentration (Figure S4) and the organic ammonium fraction (Figure S5) as a function of the time. After the concentration of the secondary species reached its maximum, the ammonium organic fraction increased with the time, indicating that the fraction of organic ammonium salt was becoming gradually important. The formation of organic




ammonium salts enhances the deprotonation of small acids, reducing their volatility and enhancing their uptake into the particle phase (Paciga et al., 2014; Wang et al., 2018; Wang and Laskin 2014). This behavior may explains the presence of small acids like formic acid ($m/z$ 47.01 ($CH_2O_2$)H$^+$) and acetic acid ($m/z$ 61.03 ($C_2H_4O_2$)H$^+$) in the particle phase. Recently Lv et al. (2022) showed that at high RH (>80%) $NH_3$ promotes the partitioning of formic and acetic acid into particulate phase through the formation of $NH_4NO_3$ and neutralization with small organic acids.

**4.3 Gas-to-particle phase partitioning**

The produced $NH_4NO_3$ particles grew to large particles and eventually tended to clog the AMS orifice over the course of the experiments. Thus, the saturation concentration calculation has been conducted for periods with particles having aerodynamic diameters below 500-600 nm, where the AMS orifice was unblocked. 1-10% of the SOA was found only in the particle phase. For the gas-to-particle phase partitioning of the SOA we used 69 ions (Table S6) that correspond to

compounds identified in previous studies (Forstner et al., 1997; Smith et al., 1998; 1999; Jang and Kamens 2001; Cocker Iii et al., 2001; Hamilton et al., 2003; 2005; Zhao et al., 2005; Huang et al., 2006; Wang et al., 2006; Sato et al., 2007; 2012; Healy et al., 2008; Wyche et al., 2009; Müller et al., 2012; Borrás and Tortajada-Genaro, 2012; Wu et al., 2014; White et al., 2014; Wu et al., 2014; Ma et al., 2018). The above 69 ions represent 41-54% of the SOA concentration and 48-55% of the secondary gas phase concentration. Figure 5 illustrates the volatility distribution in terms of O:C ratio versus logC$^*$ for the 5

experiments, while Table S6 summarizes the logC$^*$ of each ion for all five experiments. The SOA products were distributed between logC$^*$ 2 and 5 and they were found mainly in the IVOCs regime of the volatility basis set framework (Donahue et al., 2012). The volatility of the species measured during experiment #1 (cold urban emissions, Figure 5a) was centered at logC$^*$= 4 and it was the highest among all experiments most probably due to the higher total mass concentration (approximately 176 µg m$^{-3}$) formed. The volatility distributions of experiments #2 and #3 (cold urban emissions, Figures 5b

and c) were both centered at logC$^*$= 3.8 and they were alike to each other, probably because the total mass concentrations were similar (50-69 µg m$^{-3}$); they were slightly lower compared to the volatility of experiment #1 but wider, likely because of the lower SOA mass concentration. During experiments #4 and #5 (hot urban and motorway cycle emissions, Figures 5d and 5e) the formed SOA exhibited the lowest volatility distribution (average logC$^*$ =3.2), which was again wide, ranging between logC$^*$ 2.5 and 4. This behavior could be explained by the relatively lower total mass concentration (38 µg m$^{-3}$ in

experiment #5). However, the total mass concentration in experiment #4 was similar to experiment #1, while the SOA mass concentration was close to those of experiments #2 and #3, indicating that any differences in SOA and secondary gas phase distributions has a non-negligible effect on the volatility distribution.

In order to compare our volatility results with those of previous studies that implemented CHARON, we used the average volatility distribution of experiments #3 and 5, since they had SOA mass concentration closer to the ambient OA levels.

Comparing to the toluene photo-oxidation SOA volatility distribution reported by Lannuque et al. (2023) we found that the SOA products in our work had similar average O:C ratio (0.62) with respect to toluene SOA (0.55), however they were 1





order of magnitude more volatile (Figure 6). This could be explained by the lower SOA and inorganic seeds mass concentrations in the toluene SOA study (4.1±2.7 µg m$^{-3}$ and 6.6±2.7 µg m$^{-3}$ correspondingly), which probably shifted the SOA products to lower volatility. Comparing to the biogenic SOA volatility distribution by Gkatzelis et al. (2018), the

chemical species found in the gasoline emissions SOA were approximately 1-1.5 orders of magnitude more volatile and at the same time they had higher average O:C ratio (0.62) compared to the biogenic SOA (0.32) (Figure 6). This can be explained by the different chemical composition of the two types of SOA rather than the SOA levels (the biogenic SOA concentration ranged between 50-130 µg m$^{-3}$ without any inorganic seeds present, and thus the total particle concentration was higher compared to ours). Biogenic SOA contains a high fraction of acids such as nor-pinonic acid, pinonic acid, 2,2-

dimethylcyclobutane-1,3-dicarboxylic acid, pinic acid, pinalic-3-acid, 4-hydroxypinalic-3-acid (Jang and Kamens 1999; Jaoui and Kamens 2002), which generally have low volatility, while aromatic SOA was characterized by a higher fraction of aldehydes, ketones and some acids as described earlier (Tables 4 and S5).

We further selected 11 out of the 69 "parent *m/z's*" and we computed a comparison to the theoretical logC$^*$ (Figure 7). Aldehydes functional groups as glyoxal, methylglyoxal and crotonaldehyde showed lower experimental logC$^*$ in comparison

to the theoretical ones, implying that a larger fraction of these compounds partition in the particle phase than the theoretical models predict. All the other compounds had higher experimental logC$^*$ with respect to the theoretical one, suggesting that a higher-than-expected fraction of these compounds reside in the gas phase.

A tentative explanation for this behavior is the water content in the particles. Experiments were carried out at ~50% RH; during SOA formation an important amount of ammonium nitrate was formed, which is well known to take up water even at

low RH (Seinfeld and Pandis, 2016). Wet particles may control the partitioning between the gas and particle phase as water facilitates the transfer from gas to particle phase of water-soluble species such as glyoxal, crotonaldehyde, methylglyoxal which are quite water soluble with solubilities equal or higher than 100g/L. Jang and Kamens (2001) reported that products from toluene photooxidation partitioned to the condensed phase even though theoretical estimations predicted that they should be in the gas phase. The same authors found that the experimental partitioning values of aldehydes were much higher

than the predicted ones. One way to explain how relatively high vapor pressure compounds ended up in the aerosol phase is via heterogeneous reaction in the particle phase e.g., hydration, hemiacetal/ketal reactions, polymerization that form low vapour pressure products. Diols are known to have lower vapor pressure than the parent aldehydes. Another explanation is the oligomerization of small carbonyls and dicarbonyls, such as methylglyoxal and glyoxal in the particle phase that may take place (Kalberer et al., 2004; Altieri et al., 2008; Healy et al., 2008). These oligomers would fragment in the drift tube,

and they will be detected as monomers. Furthermore, the high amount of ammonium in the particle phase could catalyze heterogeneous reactions of aldehydes (e.g., Noziere et al., 2009) or enhance their partitioning in the particulate phase (Kampf et al., 2014; Ortiz-Montalvo et al., 2014).

These volatile compounds have been found in the condensed phase in many studies. For example, Volkamer et al. (2009), observed glyoxal into the particle phase in the formed SOA at 20-60% RH, while Galloway et al. (2009) found that



ammonium sulfate particles uptakes glyoxal at 40-60% RH. Methylglyoxal has been identified in both gas and particle phase of the secondary products of anthropogenic and biogenic precursors, for example in the SOA products of TMB, limonene and isoprene both in the field and the laboratory (Healy et al., 2008; Rossignol et al., 2012; 2016). Figure 8 compares the measured $logC^*$ from the above studies to the average $logC^*$ of methylglyoxal found in this work. Even though all these studies were conducted at relatively high RH (50-80%) and at similar temperatures (20-24°C) the difference between the

measured methylglyoxal $logC^*$ can be of 3.5 orders of magnitude. The major reason for this wide range is probably the total aerosol concentration in each system as for the isoprene and TMB SOA experiments it was 200-450 µg m$^{-3}$ and 100-350 µg m$^{-3}$ respectively, while for the limonene and the biogenic SOA studies it was quite lower (45 µg m$^{-3}$ and 17 µg m$^{-3}$ respectively). In our case the total aerosol concentration ranged between 40 and 180 µg m$^{-3}$, (103±69 µg m$^{-3}$) but the methylglyoxal $logC^*$ was somehow higher and more similar to the isoprene and TMB SOA systems. One explanation for this

behavior could be the oligomerization process. Kenseth et al. (2018) proposed that the synergistic $O_3$ and OH oxidation pathway during β-pinene ozonolysis leads to the formation of oligomers, while the OH photo-oxidation of β-pinene does not. Thus, methyglyoxal amount in the particle phase could be enhanced by via oligomerization processes if the formation mechanism is via ozonolysis.

   All the rest selected compounds: 2,4-furandicarbaldehyde, nitro-cresol (and isomers), maleic anhydride, citraconic

anhydride, benzaldehyde, 4-nitrophenol, 3-nitroacetophenone, terephthaldicarboxaldehyde presented reduced water solubility 0.1-16 g/L). The presence of water in the formed SOA could hinder the transfer of the non-soluble compounds towards the wet particle phase. On the other hand, it should be mentioned that the theoretical estimations are calculated (a) for individual compounds, which means that no interactions with other species are taken into account (activity coefficients were assumed to be equal to 1), (b) without taking into consideration the water content or humidity, and (c) without

considering the hygroscopicity of the compound. This suggest that the theoretical estimations may not always simulate the real ambient conditions.

   These findings suggest that there are several important factors that determine the gas-to-particle phase partitioning such as humidity, excess ammonia, water content, the hygroscopic nature of the formed compounds, the heterogeneous catalysis reactions of particulate ammonium and the oligomerization and other reactions in the particle phase. This is in general

agreement with the results of Lannuque et al., (2023) which concluded that the partitioning between gas and particle phase is a function of the organic and the aqueous phase, as well as the interactions between compounds in the particulate phase. Thus, these factors should be integrated in theoretical simulations.

**5 Conclusions**

   In this work we studied for the first time the SOA products of gasoline vehicle emissions with the recently developed

CHARON/ PTR-ToF-MS system. The emissions were produced by a EURO 5 gasoline vehicle, which was tested for cold

urban, hot urban and motorway Artemis cycles. A large part of the fresh emissions as seen by the PTR-ToF-MS was composed of light aromatic compounds ($C_6$-$C_{10}$), while the secondary organic gas phase products were distributed between $C_1$ and $C_9$ having 1-4 atoms of oxygen, with most of them being small oxygenated $C_1$- $C_3$ species. The SOA was distributed between $C_1$ and $C_{14}$ owning 1 to 6 atoms of oxygen. Interestingly, SOA had a non-negligible amount of ON compounds (6-7%) as estimated by CHARON. Based on HR-ToF-AMS analysis the fraction of nitrate linked to organonitrates was 0.12-0.20, while ammonium linked to organoammoniums was estimated only during the motorway cycle experiment with a maximum fraction of 0.19.

The volatility distributions generally depended on the formed aerosol levels. Higher total mass concentrations (176 µg m$^{-3}$ of which 46 µg m$^{-3}$ were SOA) lead to more volatile SOA (logC$^*$=4) with a relatively narrow volatility distribution. For total mass concentration (50-69 µg m$^{-3}$ containing 9-14 µg m$^{-3}$ of SOA) the volatility was wider and was centered at logC$^*$=3.8, while for lower total mass concentrations (38 µg m$^{-3}$ with 3.9 µg m$^{-3}$ SOA) the products were less volatile with an average logC$^*$=3.2.

Using the molecular formula of the "parent" compounds we calculated the saturation concentration of individual species. We found that low water-soluble compounds and highly water-soluble compounds declined from the theoretical estimations by 1-3 orders of magnitude. We conclude that the gas-to-particle partitioning is affected not only by the concentration of the formed aerosol and the temperature, but also by other factors such as the RH level, the existence of water in the OA mixture, the hygroscopicity of the formed compounds, and possible reactions in the particulate phase (e.g., oligomerization and/or heterogeneous catalysis by particulate ammonium).

*Supplement.* The supplement related to this article is available on-line at: (link will be included by Copernicus).

*Author Contribution.* BD, EK and KS designed the research. BTR, YL and BV contributed to the experimental set-up and the experimental procedure. EK and BM performed the experiments. EK analyzed, interpreted the data and drafted the article. EK, BM, TRB, LY, VB, KS, and BD revised the article. BD, LY, and KS were responsible for funding acquisition.

*Competing interests.* The authors declare that they have no conflict of interest.

*Acknowledgements.* The authors thank the MASSALYA instrumental platform (Aix Marseille Université, lce.univ-amu.fr) for the technical support and the instrumentation calibration. The authors would like to thank Patrick Tassel and Pascal Perret for the technical support on the chassis dynamometer experiments at AME-EASE, Univ Gustave Eiffel, Lyon. The authors thank Amélie Bertrand for the development of PeTeR v3.5 toolkit.



*Financial Support.* This research has been funded by the ADEME CORTEA program with the projects MAESTRO (no. 1766C0001) and MAESTRO-EU6 (no. 1866C0001) and by the ANR program via the project POLEMICS (grant ANR-18-CE22-0011).

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



**Table 1: Experimental conditions for the 5 experiments.**

| | Exp #1 | Exp #2 | Exp #3 | Exp #4 | Exp #5 |
|---|---|---|---|---|---|
| Type of cycle (part of the cycle) | Cold Urban 1 (first 5 min) | Cold Urban 2 (whole cycle) | Cold Urban 3 (first 5 min) | Hot Urban (whole cycle) | Motorway (whole cycle) |
| Total Fresh VOCs concentration (ppb) after filling the chamber (before any dilution in chamber) | 3414 | 4238 | 4186 | 125 | 89 |
| Initial NOx (ppb) (before any dilution in chamber) | 2153 | 2600 | 2158 | 550 | 212 |
| VOC (ppbC) /$NO_x$ (ppb) | 10.1 | 10.3 | 12.4 | 1.4 | 2.8 |
| Dilution ratio of fresh VOCs (inside chamber) | 1.0 | 3.2 | 5.1 | 1.0 | 2.3 |
| Partial dilution ratio (emissions to initial fresh VOCs into chamber) | 112 | 28 | 101 | 48 | 48 |
| Total dilution ratio (emissions to fresh VOCs after chamber dilution) | 112 | 91 | 518 | 48 | 111 |
| Average OH concentration (molecules $cm^{-3}$) during the SVOC identification sample (Table 3) and CHARON sampling (Table 4) | $6.6\times10^5$ | $9.3\times10^5$ | $1.0\times10^6$ | $2.3\times10^6$ | $1.8\times10^6$ |
| OH exposure (molecules $cm^{-3}$ s) during the SVOC identification sample (Table 3) and CHARON sampling (Table 4) | $6.2\times10^9$ | $5.2\times10^9$ | $8.2\times x10^9$ | $1.6\times10^{10}$ | $1.0\times10^{10}$ |
| OH exposure (days) during the SVOC identification sample (Table 3) and CHARON sampling (Table 4)[a] | 0.1 | 0.08 | 0.13 | 0.25 | 0.16 |
| Temperature (ºC) (during CHARON sampling) | 25.8 | 22.8 | 25.5 | 22.1 | 24.5 |
| RH (%) (during CHARON sampling) | 46.5 | 53.5 | 40 | 55.4 | 39.7 |

[a]Assuming a daily average OH concentration of $1.5x10^6$ molecules $cm^{-3}$.



**Table 2: Measured accurate *m/z*, elemental composition of the detected ions and tentative assignment of fresh VOCs emitted**
**during cold urban, hot urban and motorway Artemis cycles.**

| *m/z* | Molecular formula | Possible compound(s) | % in total fresh VOC | | | | |
|---|---|---|---|---|---|---|---|
| | | | Exp #1 | Exp #2 | Exp #3 | Exp #4 | Exp #5 |
| | | | Cold Urban 1 | Cold Urban 2 | Cold Urban 3 | Hot Urban | Motorway |
| 31.02 | $(CH_2O)H^+$ | Formaldehyde | 0.5 | 0.6 | 0.4 | 0.0 | 0.0 |
| 33.03 | $(CH_3OH)H^+$ | Methanol | 0.8 | 0.4 | 0.5 | 0.5 | 1.1 |
| 45.03 | $(C_2H_4O)H^+$ | Acetaldehyde | 1.5 | 1.6 | 1.7 | 0.7 | 2.3 |
| 47.05 | $(C_2H_6O)H^+$ | Ethanol | 2.8 | 2.3 | 2.5 | 0.5 | 1.3 |
| 57.03 | $(C_3H_4O)H^+$ | Acrolein | 0.3 | 0.3 | 0.2 | 0.2 | 0.4 |
| 59.05 | $(C_3H_6O)H^+$ | Acetone | 0.4 | 0.3 | 0.6 | 0.7 | 1.9 |
| 41.04 | $(C_3H_4)H^+$ | | | | | | |
| 43.05 | $(C_3H_6)H^+$ | | | | | | |
| 57.07 | $(C_4H_8)H^+$ | Alkyl fragments | 21.7 | 25.7 | 23.3 | 21.5 | 19.8 |
| 69.07 | $(C_5H_8)H^+$ | | | | | | |
| 71.09 | $(C_5H_{10})H^+$ | | | | | | |
| 79.05 +80.06 | $(C_6H_6)H^+$ $([13C]C_5H_6)H^+$ | Benzene | 4.8 | 5.3 | 6.2 | 13.2 | 7.1 |
| 93.07 +94.07 | $(C_7H_8)H^+$ $([13C]C_6H_8)H^+$ | Toluene | 9.6 | 9.6 | 9.5 | 11.7 | 4.6 |
| 107.09 +108.09 | $(C_8H_{10})H^+$ $([13C]C_7H_{10})H^+$ | $C_8$ aromatics | 26.1 | 24.4 | 26.3 | 16.4 | 10.9 |
| 121.10 +122.11 | $(C_9H_{12})H^+$ $([13C]C_8H_{12})H^+$ | $C_9$ aromatics | 14.0 | 13.8 | 14.3 | 8.3 | 12.3 |
| 135.12 +136.12 | $(C_{10}H_{14})H^+$ $([13C]C_9H_{14})H^+$ | $C_{10}$ aromatics | 1.1 | 1.4 | 1.2 | 1.8 | 3.4 |
| | **Fraction of the above compounds to the total fresh VOC** | | **0.84** | **0.86** | **0.87** | **0.75** | **0.65** |





**Table 3:** Measured accurate *m/z*'s, elemental composition of the detected ions and tentative assignment of the most important secondary gas phase products produced from cold urban, hot urban, and motorway Artemis cycles emissions. Assignment was supported by literature studies of individual aromatic compounds (Forstner et al., 1997; Smith et al., 1998; 1999; Jang and Kamens 2001; Cocker Iii et al., 2001; Hamilton et al., 2003; 2005; Zhao et al., 2005; Huang et al., 2006; Wang et al., 2006; Sato et al., 2007; 2012; Healy et al., 2008; Wyche et al., 2009; Müller et al., 2012; Borrás and Tortajada-Genaro, 2012; White et al., 2014; Wu et al., 2014; Ma et al., 2018; Schwantes et al., 2017).

| *m/z* | Molecular formula | Possible compound(s) | Concentration (ppb) | | | | |
| --- | --- | --- | --- | --- | --- | --- | --- |
| | | | Exp #1 | Exp #2 | Exp #3 | Exp #4 | Exp #5 |
| | | | Cold Urban 1 | Cold Urban 2 | Cold Urban 3 | Hot Urban | Motorway |
| 31.02 | $(CH_2O)H^+$ | Formaldehyde | 62.0 | 17.9 | 16.9 | 2.2 | 1.1 |
| 33.03 | $(CH_3OH)H^+$ | Methanol | 5.9 | 1.7 | 8.7 | 1.8 | 1.8 |
| 45.03 +63.04 | $(C_2H_4O)H^+$ $+(C_2H_4O)(H_2O)H^+$ | Acetaldehyde and its hydrate | 190.8 | 69.0 | 74.2 | 12.5 | 4.8 |
| 46.03 | $(CH_3NO)H^+$ | **Formamide/Nitromethane** | 12.4 | - | 6.2 | 2.5 | 1.0 |
| 47.01 +48.02 +65.02 | $(CH_2O_2)H^+$ $[13C][H_2O_2]H^+$ $+H_2O(CH_2O_2)H^+$ | Formic acid / Formic acid isotope | 18.6 | 2.3 | 61.6 | 5.0 | - |
| 57.03 | $(C_3H_4O)H^+$ | Acrolein/ Hydroxy acetone fragment | 19.8 | 6.7 | 7.6 | 1.3 | 0.8 |
| 59.01 | $(C_2H_2O_2)H^+$ | Glyoxal | 3.5 | - | - | - | 0.1 |
| 59.05 | $(C_3H_6O)H^+$ | Acetone | 185.3 | 66.6 | 74.3 | 18.4 | 7.7 |
| 61.03 +43.02 +79.04 | $(C_2H_4O_2)H^+$ $(C_2H_2O)H^+$ $((C_2H_4O_2)H_2O)H^+$ | Hydroxy acetaldehyde/Acetic acid / Acetic acid fragment/Hydroxy acetaldehyde fragment and hydrate | 185.4 | 31.5 | 143.0 | 32.0 | 15.5 |
| 71.05 | $(C_4H_6O)H^+$ | Butenal, Crotonaldehyde, Methacrolein, MVK | 12.7 | 4.1 | 2.5 | 0.8 | 0.6 |
| 73.03 +74.03 (13% of 73.03) | $(C_3H_4O_2)H^+$ $([13C][C_2H_4O_2]H^+$ $(C_2H_4O)H^+$ | Methylglyoxal / Methylglyoxal isotope / Methylglyoxal fragment[*] | 109.3 | 32.2 | 41.9 | 5.3 | 2.3 |
| 73.06 | $(C_4H_8O)H^+$ | Butanone | 42.1 | 13.4 | 13.7 | 3.1 | 1.1 |
| 75.04 | $(C_3H_6O_2)H^+$ | Hydroxy acetone/Propanoic acid | 18.5 | 3.7 | 12.8 | 2.0 | 1.5 |
| 77.02 | $(C_2H_4O_3)H^+$ | PAN fragment | 10.0 | 1.0 | 21.5 | 1.2 | 0.5 |
| 83.05 | $(C_5H_6O)H^+$ | 4-oxo-2-pentanal | 2.7 | 0.6 | - | - | - |

950



| m/z | Formula | Assignment | | | | | |
|---|---|---|---|---|---|---|---|
| 85.03 | (C4H4O2)H+ | Butenedial/ Furanone | 8.5 | 1.2 | 1.4 | 0.4 | 0.2 |
| 85.06 | (C5H8O)H+ | Methyl butenal | 6.5 | 1.8 | 1.9 | 0.62 | 0.3 |
| 87.04 | (C4H6O2)H+ | Butanedionediacetyl/Oxo butanal/2,3-Epoxy-butandial/3-hydroxybutenone/crotonic acid/butanedial | 31.2 | 8.9 | 13.6 | 2.7 | 1.0 |
| 87.08 | (C5H10O)H+ | Pentanone | 16.3 | 5.1 | 5.9 | 1.3 | 0.4 |
| 89.02 | (C3H4O3)H+ | Methyl glyoxylic acid | 6.5 | 0.6 | 2.1 | 0.7 | 0.4 |
| 89.06 | (C4H8O2)H+ | Hydroxy butanone/butanoic acid | 4.4 | 0.7 | 3.1 | 1.1 | 0.7 |
| 95.05 | (C6H6O)H+ | Phenol | 2.3 | 0.6 | - | 0.2 | - |
| 97.03 | (C5H4O2)H+ | Furfural/ 4-oxo-2,3-pentanedial | 8.5 | 0.7 | 2.0 | 0.3 | 0.1 |
| 99.01 | (C4H2O3)H+ | Maleic anhydride | 12.9 | 1.9 | 6.5 | 0.8 | 0.9 |
| 99.04 | (C5H6O2)H+ | 4-oxo-pentenal/methylbutenedial/ Methylfuranone | 23.9 | 4.9 | 3.6 | 0.7 | 0.3 |
| 99.08 | (C6H10O)H+ | No literature data | 5.6 | 1.7 | 1.7 | 0.6 | 0.2 |
| 101.02 / 83.01 | (C4H4O3)H+ / (C4H2O2)H+ | Succinic anhydride/dioxobutanal/ 4-oxo-butenoic acid/ Furanone | 8.0 | 0.8 | 3.1 | 0.9 | 0.7 |
| 101.06 | (C5H8O2)H+ | Dihydro-methyl-furanone/ Pentanedione | 8.6 | 2.1 | 3.3 | 0.8 | 0.4 |
| 103.04 | (C4H6O3)H+ | Hydroxy-oxobutanal | 6.0 | - | 1.3 | 0.3 | 0.1 |
| 107.05 | (C7H6O)H+ | Benzaldehyde | 7.4 | 0.6 | 2.0 | - | - |
| 109.07 | (C7H8O)H+ | Cresols/Benzyl alcohol | 3.1 | 1.3 | - | - | - |
| 111.04 | (C6H6O2)H+ | Methylfuraldehyde (isomers)/ Cyclohexenedione/ Dihydroxybenzene (isomers) | 11.4 | 1.0 | 1.9 | 0.2 | - |
| 113.02 | (C5H4O3)H+ | Methylfurandione | 25.2 | 3.4 | 8.8 | 1.0 | 0.7 |
| 113.06 | (C6H8O2)H+ | Methyl-oxo-pentenal/Ethyl-furanone/ Dimethyl-furanone (possible interference with phenol m/z 95.05) | 29.0 | 8.5 | 3.6 | 0.7 | 0.3 |
| 113.10 | (C7H12O)H+ | unknown | 5.1 | 1.3 | 1.2 | 0.6 | 0.1 |
| 115.04 | (C5H6O3)H+ | 4-oxo-pentenoic acid | 11.6 | 1.2 | 3.6 | 0.6 | 0.4 |



| m/z | Ion | Compound | | | | | |
|---|---|---|---|---|---|---|---|
| | | /Hydroxy methyl dicarbonyl butene (interference with furfural at m/z 97.03) | | | | | |
| 115.08 | $(C_6H_{10}O_2)H^+$ | Dimethyl furanone | 4.1 | 1.2 | 2.1 | 0.6 | 0.2 |
| 117.02 | $(C_4H_4O_4)H^+$ | Maleic acid | 2.3 | 0.2 | 0.5 | 0.1 | 0.1 |
| 121.07 | $(C_8H_8O)H^+$ | Acetophenone/ m-Toluraldehyde | 10.9 | 2.7 | 3.8 | 0.2 | - |
| 127.04 | $(C_6H_6O_3)H^+$ | Hydroxyquinol/Hydroxymethylfurfural (possible interference with Benzoquinone/Hydroquinone at m/z 109.03) | 10.3 | 1.2 | 3.1 | 0.4 | 0.3 |
| 129.06 | $(C_6H_8O_3)H^+$ | Methyl−4−oxo−2−pentenoic acid /Hydroxy-oxo-hexenal/Furanone | 7.6 | 0.8 | 2.4 | - | 0.1 |
| 138.06 | $(C_7H_7NO_2)H^+$ | **Nitrotoluene** | 3.8 | 0.6 | - | - | - |
| 139.04 | $(C_7H_8O_3)H^+$ | Methyl-cyclohexene tricarbonyl/ Hydroxy methyl benzoquinone | 4.8 | 0.6 | 1.1 | 0.3 | 0.1 |
| 140.03 | $(C_6H_5NO_3)H^+$ | **Nitrophenol** | 3.8 | 0.5 | 0.7 | 0.3 | 0.1 |
| 141.06 | $(C_7H_8O_3)H^+$ | Oxo-heptedienoic acid/ epoxy-methyl-hexenedial (interference with Hydroxybenzaldehyde/Benzoic acid/Methylbenzoquinone at m/z 123.046) | 6.3 | 0.6 | 1.7 | 0.3 | 0.1 |
| 154.05 | $(C_7H_7NO_3)H^+$ | **Nitrocresol** | 4.2 | 0.4 | 1.2 | 0.3 | 0.1 |
| 168.07 | $(C_8H_9NO_3)H^+$ | **Ethyl-nitrophenol/ Dimethyl-nitrophenol** | 6.4 | 0.6 | 1.8 | 0.2 | - |
| | | **Total secondary VOC concentration (ppb)** | 1419.9 | 334.0 | 611.0 | 115.1 | 52.3 |
| | | **Fraction of the above compounds to the total secondary VOC** | 0.83 | 0.92 | 0.94 | 0.92 | 0.90 |



**Table 4: Measured accurate m/z's, elemental composition of the detected ions and tentative assignment of the most important SOA products produced from cold urban, hot urban, and motorway Artemis cycles emissions and identified using CHARON. Assignment was supported by literature studies of individual aromatic compounds (Forstner et al., 1997; Smith et al., 1998; 1999; Jang and Kamens 2001; Cocker Iii et al., 2001; Hamilton et al., 2003; 2005; Zhao et al., 2005; Huang et al., 2006; Wang et al., 2006; Sato et al., 2007; 2012; Healy et al., 2008; Wyche et al., 2009; Müller et al., 2012; Borrás and Tortajada-Genaro, 2012; White et al., 2014; Wu et al., 2014; Ma et al., 2018; Schwantes et al., 2017).**

| m/z | Molecular formula | Possible compound(s) | Concentration (ppb) | | | | |
| --- | --- | --- | --- | --- | --- | --- | --- |
| | | | Exp #1 Cold Urban 1 | Exp #2 Cold Urban 2 | Exp #3 Cold Urban 3 | Exp #4 Hot Urban | Exp #5 Motorway |
| 45.03 +63.04 | $(C_2H_4O)H^+$ +$(C_2H_4O)(H_2O)H^+$ | Fragments from larger compounds | 29.4 | 2.0 | 6.5 | 3.6 | - |
| 47.01 +48.02 +65.02 | $(CH_2O_2)H^+$ $[13C]H_2O_2)H^+$ +$H_2O(CH_2O_2)H^+$ | Formic acid Formic acid isotope and its hydrate | 74.4 | 7.1 | 14.5 | 10.5 | 0.1 |
| 57.03 | $(C_3H_4O)H^+$ | Methyl-propenal | 13.1 | 0.6 | 2.4 | 0.8 | 0.2 |
| 59.01 | $(C_2H_2O_2)H^+$ | Glyoxal | 5.6 | - | - | 0.9 | 0.1 |
| 59.05 | $(C_3H_6O)H^+$ | Acetone | 30.7 | 8.1 | 8.4 | 11.8 | - |
| 61.03 +43.02 +79.04 | $(C_2H_4O_2)H^+$ $(C_2H_2O)H^+$ $((C_2H_4O_2)H_2O)H^+$ | Hydroxy acetaldehyde/Acetic acid Acetic acid fragment/Hydroxy acetaldehyde fragment and hydrate | 131.3 | 11.7 | 44.6 | 18.7 | 1.2 |
| 71.01 | $(C_3H_2O_2)H^+$ | No literature data | 6.3 | 0.3 | 1.0 | 0.5 | - |
| 73.03 +74.03 (13% of 73.03) | $(C_3H_4O_2)H^+$ $([13C]C_2H_4O_2)H^+$ | Methylglyoxal Methylglyoxal isotope | | | | | |
| | $(C_2H_4O)H^+$ | Methylglyoxal fragment* | 43.9 | 3.7 | 14.5 | 5.6 | 0.7 |
| 74.02 | $(C_2H_3NO_2)H^+$ | No literature data | 8.1 | 0.6 | 1.9 | 1.4 | - |
| 75.01 | $(C_2H_2O_3)H^+$ | Glyoxylic acid | 2.2 | 0.2 | 0.6 | 0.3 | 0.0 |
| 75.04 | $(C_3H_6O_2)H^+$ | Hydroxy acetone/Propanoic acid | 15.7 | 1.3 | 5.2 | 1.7 | 0.3 |
| 83.05 | $(C_5H_5O)H^+$ | 4-oxo-2-pentanal | 5.5 | 0.2 | 0.9 | 0.4 | 0.1 |
| 85.03 | $(C_4H_4O_2)H^+$ | Butenedial/Furanone | 16.1 | 0.8 | 2.9 | 1.2 | 0.2 |
| 87.04 | $(C_4H_6O_2)H^+$ | Butanedionediacetyl/Oxo butanal/ Epoxy-butandial/ Hydroxybutenone/Crotonic acid/ butanedial | 22.9 | 1.5 | 6.8 | 2.5 | 0.4 |
| 89.02 | $(C_3H_4O_3)H^+$ | Methyl glyoxylic acid/ Hydroxy-pronadial | 18.2 | 1.0 | 4.4 | 3.3 | 0.3 |
| 91.04 | $(C_3H_6O_3)H^+$ | Lactic acid | 4.2 | 0.3 | 1.1 | 0.3 | - |

955





| m/z | Ion formula | Compound | | | | | |
|---|---|---|---|---|---|---|---|
| 95.05 | (C6H6O)H+ | Phenol | 3.4 | 0.1 | 0.2 | 0.0 | 0.1 |
| 97.03 | (C5H4O2)H+ | Furfural/ 4-oxo-pentanedial | 10.8 | 0.6 | 1.7 | 0.9 | - |
| 97.06 | (C6H8O)H+ | Dimethylfuran | 7.5 | 0.3 | 1.0 | 0.4 | 0.0 |
| 99.01 | (C4H2O3)H+ | Maleic anhydride | 21.3 | 1.7 | 1.6 | 1.9 | 0.5 |
| 99.04 | (C5H6O2)H+ | 4-oxo-pentenal/Methylbutenedial/ Methylfuranone | 20.3 | 1.1 | 4.2 | 1.6 | 0.3 |
| 101.02 | (C4H4O3)H+ | Succinic anhydride/ Dioxobutanal/ 4-oxo-butenoic acid/ Furanone | 18.5 | 1.3 | 3.9 | 2.0 | 0.3 |
| 101.06 | (C5H8O2)H+ | Dihydro-methyl-furanone/ Pentanedione | 6.9 | 0.4 | 2.3 | 0.9 | 0.2 |
| 103.04 | (C4H6O3)H+ | Hydroxy-oxobutanal + isomers | 14.0 | 0.7 | 4.0 | 1.3 | 0.2 |
| 105.02 | (C3H4O4)H+ | Malonic acid | 7.4 | 0.4 | 2.1 | 0.6 | 0.2 |
| 107.05 | (C7H6O)H+ | Benzaldehyde | 6.4 | 0.1 | 0.4 | 0.1 | 0.1 |
| 111.04 | (C6H6O2)H+ | Methylfuraldehyde (isomers)/ Cyclohexenedione/ Dihydroxybenzene (isomers) | 18.0 | 0.7 | 2.6 | 1.0 | 0.1 |
| 113.02 | (C5H4O3)H+ | Methylfurandione | 24.9 | 1.9 | 4.1 | 1.8 | 0.4 |
| 113.06 | (C6H8O2)H+ | Methyl-oxo-pentenal/Ethyl-furanone/ Dimethyl-furanone (possible interference with phenol m/z 95.05) | 13.6 | 0.7 | 3.0 | 1.0 | 0.1 |
| 115.04 | (C5H6O3)H+ | 4-oxo-pentenoic acid /Hydroxy methyl dicarbonyl butene (interference with furfural at m/z 97.03) | 26.6 | 2.1 | 7.7 | 2.4 | 0.3 |
| 117.02 | (C4H4O4)H+ | Maleic acid | 13.5 | 0.6 | 1.3 | 0.7 | 0.1 |
| 117.06 | (C5H8O3)H+ | Hydroxy-pentanedione/ 4-Oxo-pentanoic acid | 6.7 | 0.5 | 2.6 | 0.9 | 0.2 |
| 125.06 | (C7H8O2)H+ | Oxo-heptanadienal(isomers)/ Methyl catechol/Acetyl-methyl-furan/Ethyl-furaldehyde | 11.5 | 0.5 | 2.2 | 0.6 | 0.1 |
| 127.04 | (C6H6O3)H+ | Hydroxyquinol/Hydroxymethylfurfural (possible interference with Benzoquinone/Hydroquinone at m/z 109.03) | 18.5 | 0.9 | 4.6 | 1.4 | 0.2 |
| 129.06 | (C6H8O3)H+ | Methyl-4-oxo-2-pentenoic acid /Hydroxy-oxo-hexenal/Furanone | 15.0 | 0.9 | 4.3 | 1.2 | 0.2 |
| 131.03 | (C5H6O4)H+ | Cicatronic acid/ Dioxopentanoic | 8.4 | 0.4 | 2.0 | 0.7 | 0.1 |





| m/z | Formula | Compound | | | | | |
|---|---|---|---|---|---|---|---|
| | | acid (isomers) | | | | | |
| 138.05 | $(C_7H_7NO_2)H^+$ | **Nitrotoluene** | 3.2 | 0.1 | 0.3 | 0.1 | 0.1 |
| 139.04 | $(C_7H_6O_3)H^+$ | Methyl-cyclohexene tricarbonyl/ Hydroxy methyl benzoquinone | 6.8 | 0.4 | 1.6 | 0.6 | 0.1 |
| 139.08 | $(C_8H_{10}O_2)H^+$ | Dimethyl-benzenediol/ Methyl-oxo-heptadienal (isomers) | 8.4 | 0.3 | 1.0 | 0.3 | - |
| 140.03 | $(C_6H_5NO_3)H^+$ | **Nitrophenol** | 4.1 | 0.2 | 0.7 | 0.4 | 0.1 |
| 141.06 | $(C_7H_8O_3)H^+$ | Oxo-heptedienoic acid/ epoxy-methyl-hexenedial (interference with Hydroxybenzaldehyde/Benzoic acid/Methylbenzoquinone at m/z 123.046) | 16.1 | 0.7 | 3.9 | 1.1 | 0.2 |
| 143.03 | $(C_6H_6O_4)H^+$ | Muconic acid/ Dihydroxy-methyl-pyranone (isomers) | 8.1 | 0.4 | 2.1 | 0.8 | 0.1 |
| 145.05 | $(C_6H_8O_4)H^+$ | Carbonyl hydroxy methyl butene carboxylic acid (isomers)/ Methyl-hydroxy-dioxo-pentanal | 6.5 | 0.3 | 1.7 | 0.5 | 0.1 |
| 153.06 | $(C_8H_8O_3)H^+$ | Hydroxy-dimethyl-cyclohexadiene-dione | 10.4 | 0.4 | 2.0 | 0.5 | 0.1 |
| 154.05 | $(C_7H_7NO_3)H^+$ | **Nitrocresol** | 4.9 | 0.2 | 0.8 | 0.3 | 0.1 |
| 155.07 | $(C_8H_{10}O_3)H^+$ | Methyl-heptene-trione | 10.2 | 0.3 | 1.8 | 0.5 | 0.1 |
| 156.03 | $(C_6H_5NO_4)H^+$ | **Nitrocatechol** | 4.2 | 0.1 | 0.5 | 0.3 | - |
| 157.05 | $(C_7H_8O_4)H^+$ | Hydroxy-dioxo-heptenal (isomers)/Hydroxy methyl trioxo cyclohexene/Tetrahydroxy toluene (interference with m/z 139.04) | 9.6 | 0.3 | 2.5 | 0.6 | 0.1 |
| 168.07 | $(C_8H_9NO_3)H^+$ | **Ethyl-nitrophenol/ Dimethyl-nitrophenol** | 3.7 | 0.1 | 0.4 | - | 0.1 |
| 170.05 | $(C_7H_7NO_4)H^+$ | **Methylnitrocatechol** | 6.8 | 0.2 | 0.6 | - | - |
| 171.07 | $(C_8H_{10}O_4)H^+$ | No literature data | 11.4 | 0.3 | 2.2 | 0.5 | 0.1 |
| 173.04 | $(C_7H_8O_5)H^+$ | Pentahydroxy toluene | 7.9 | 0.2 | 1.1 | 0.3 | - |
| | | **Total SOA concentration (ppb)** | 1272.0 | 91.9 | 316.1 | 156.3 | 24.4 |
| | | **Fraction of the above compounds to the total SOA** | 0.65 | 0.65 | 0.62 | 0.59 | 0.35 |



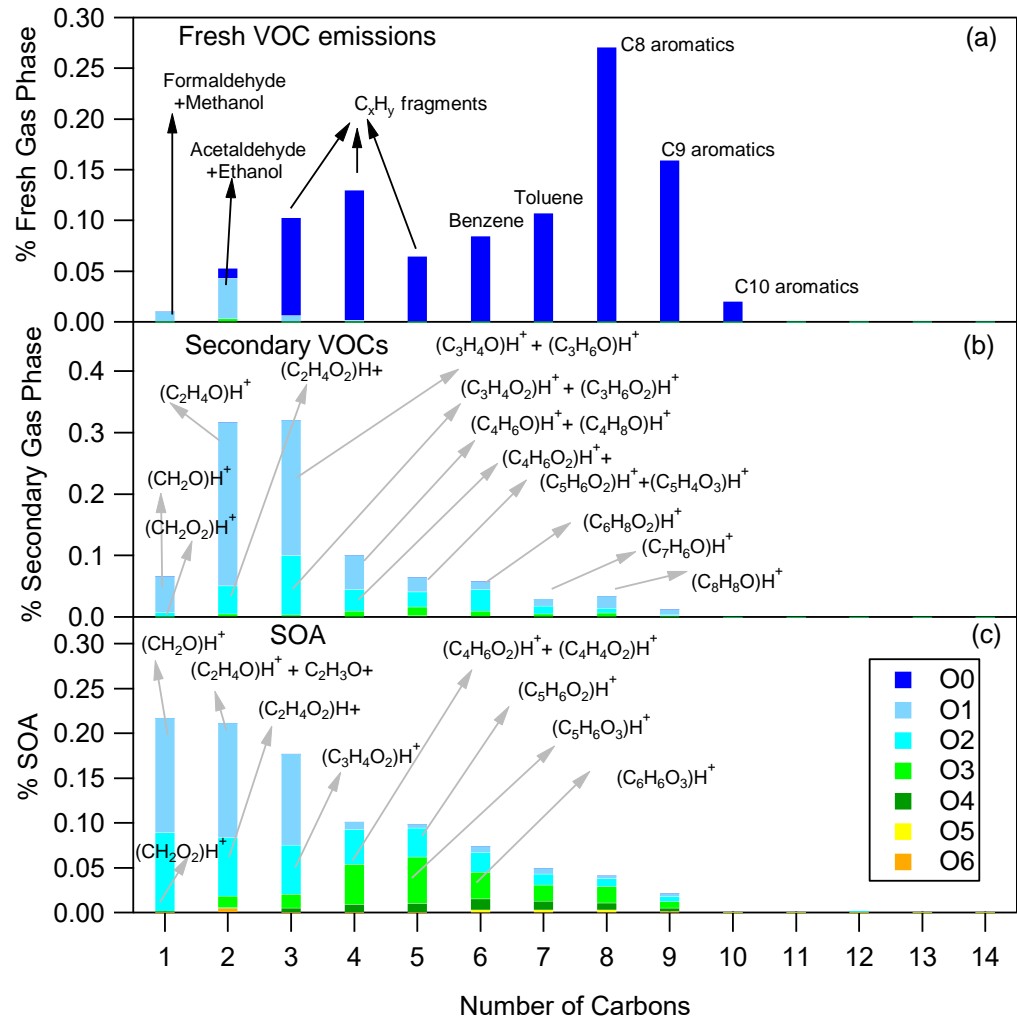

**Figure 1: Fresh VOCs (a), secondary VOCs (b) and SOA (c) distributions for cold urban Artemis emissions (experiment #2). The secondary VOC and SOA distributions correspond to an OH exposure equals to $5.2 \times 10^9$ molecules cm$^{-3}$ s**



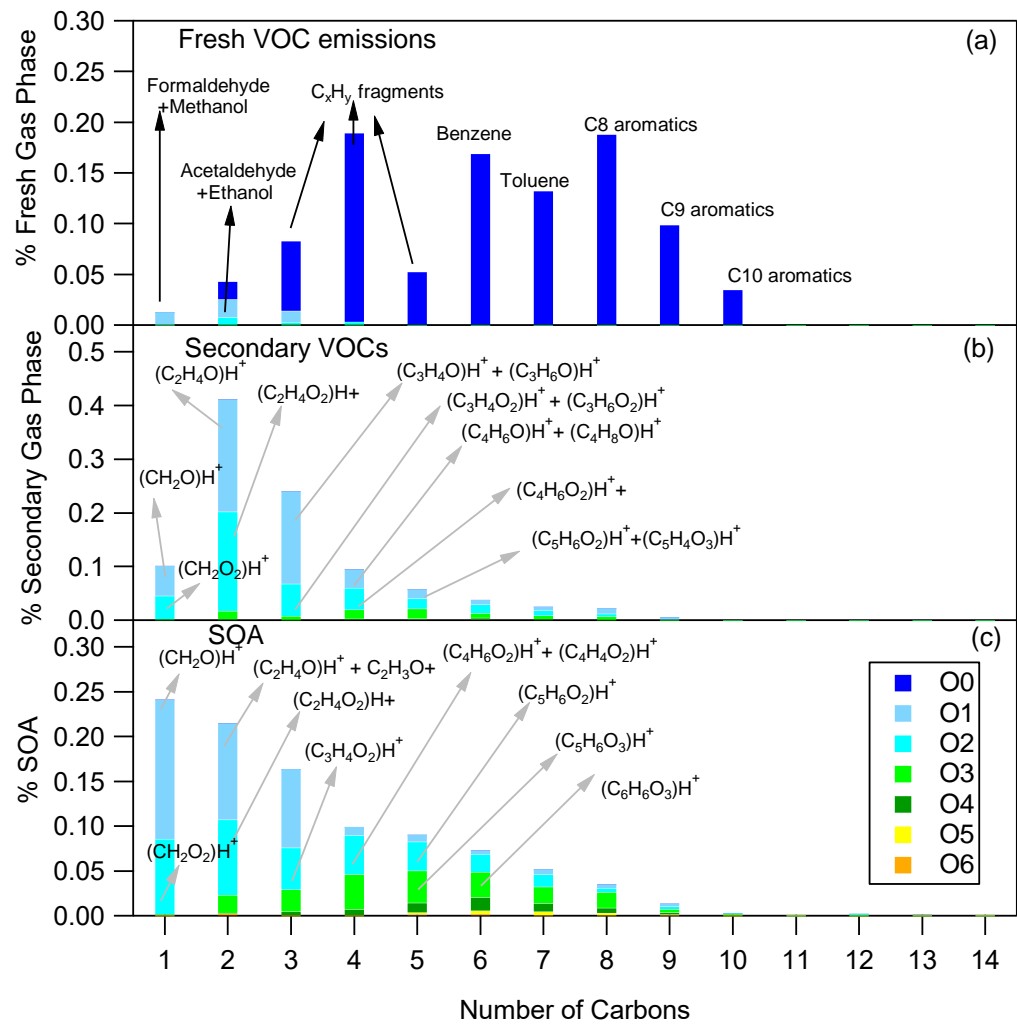

**Figure 2: Fresh VOCs (a), secondary VOCs (b) and SOA (c) distributions for hot urban Artemis emissions (experiment #4). The secondary VOC and SOA distributions correspond to an OH exposure equals to $1.6 \times 10^{10}$ molecules cm$^{-3}$ s.**





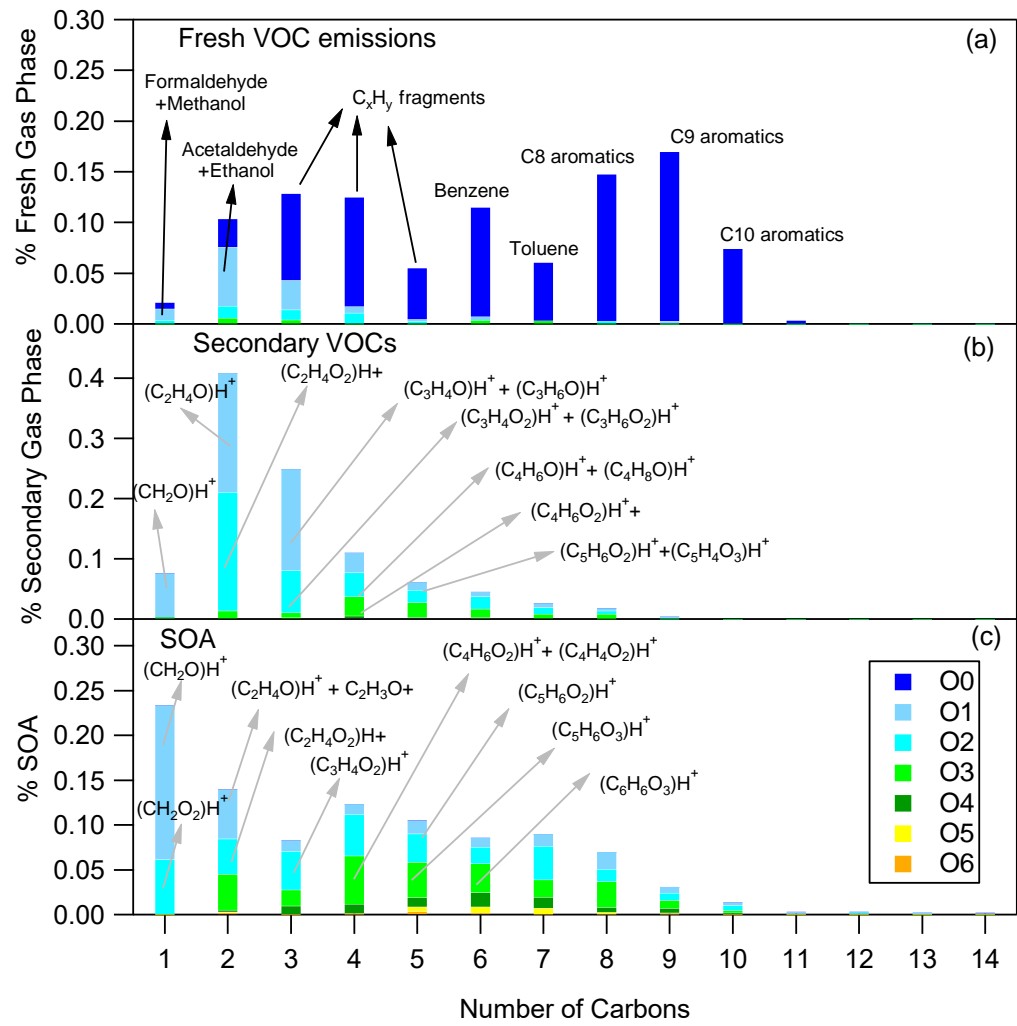

**Figure 3: Fresh VOCs (a), secondary VOCs (b) and SOA (c) distributions for motorway Artemis emissions (experiment #5). The secondary VOC and SOA distributions correspond to an OH exposure equals to $1.0 \times 10^{10}$ molecules cm$^{-3}$ s.**



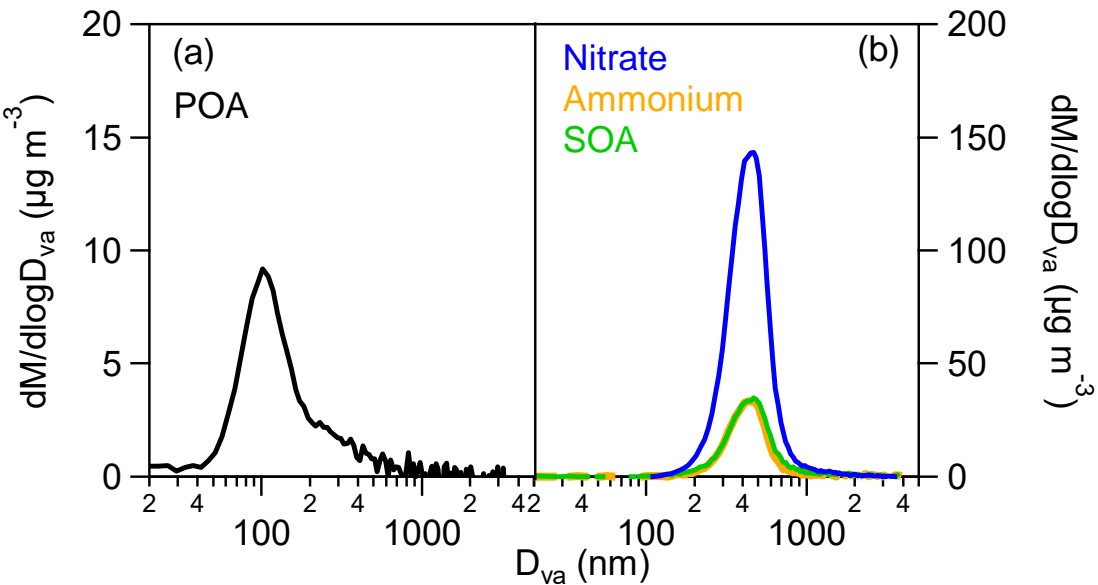

**Figure 4: AMS mass distributions versus the aerodynamic diameter of (a) the primary emitted organic aerosol (POA) and (b) the produced SOA, ammonium and nitrate at the time when the CHARON sample, during the experiment 2 (cold urban Artemis emissions). The fresh organic emissions have a peak at 100 nm, while the secondary products at 450 nm.**



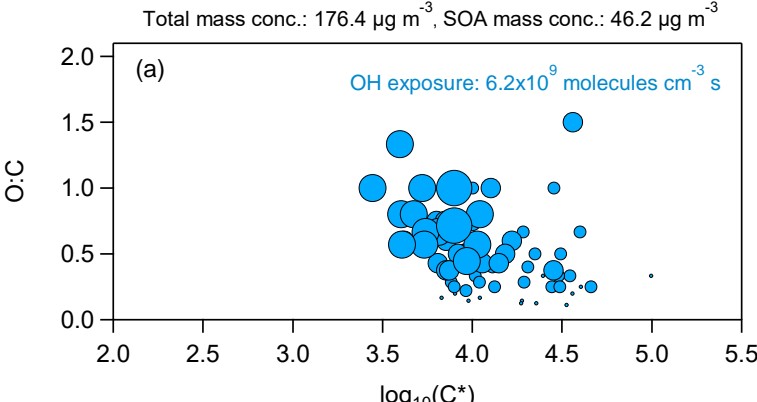

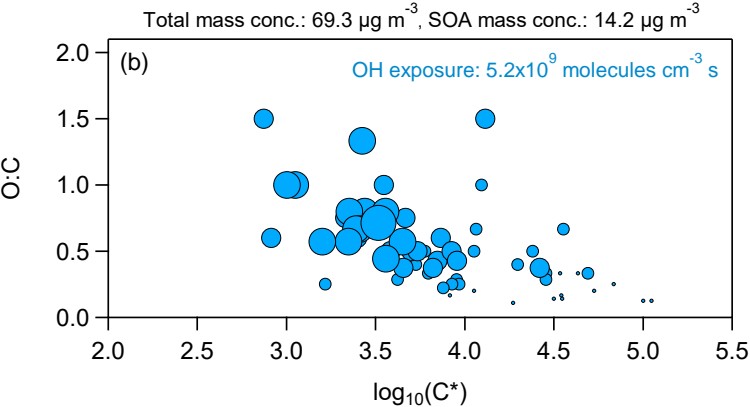

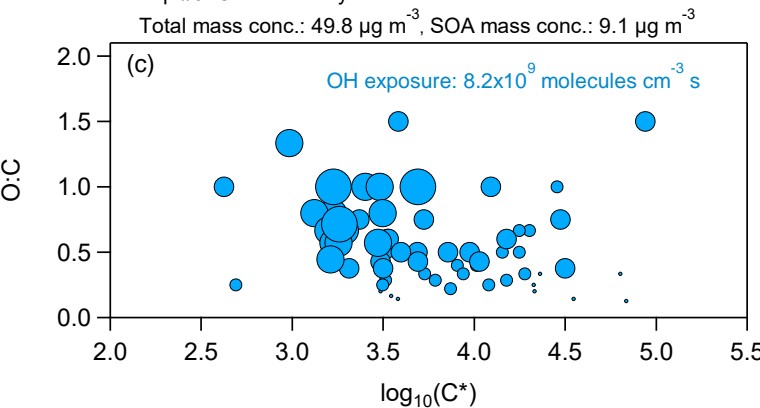



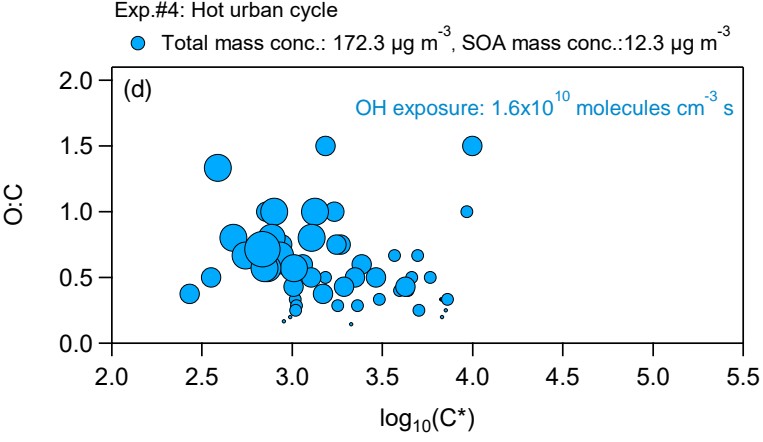

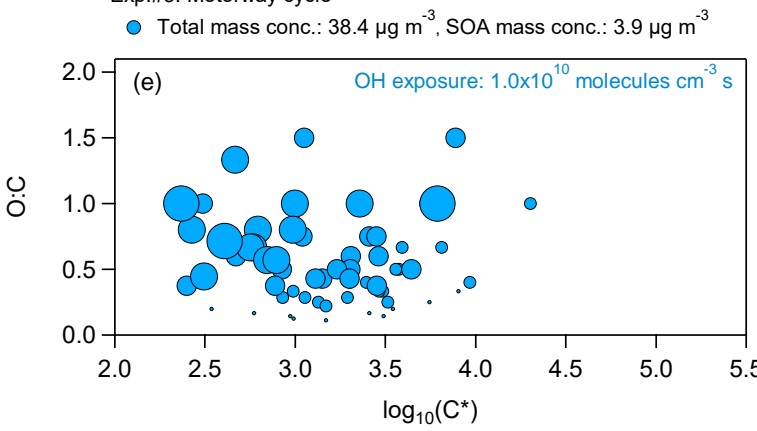

**Figure 5: Volatility distribution in terms of O:C ratio versus logC* for the 5 experiments with the different cycle exhausts. The different size of the circles represents the atoms of oxygen in the compound: the smaller size corresponds to 1 atom of O, while the largest size to 5 atoms of O.**



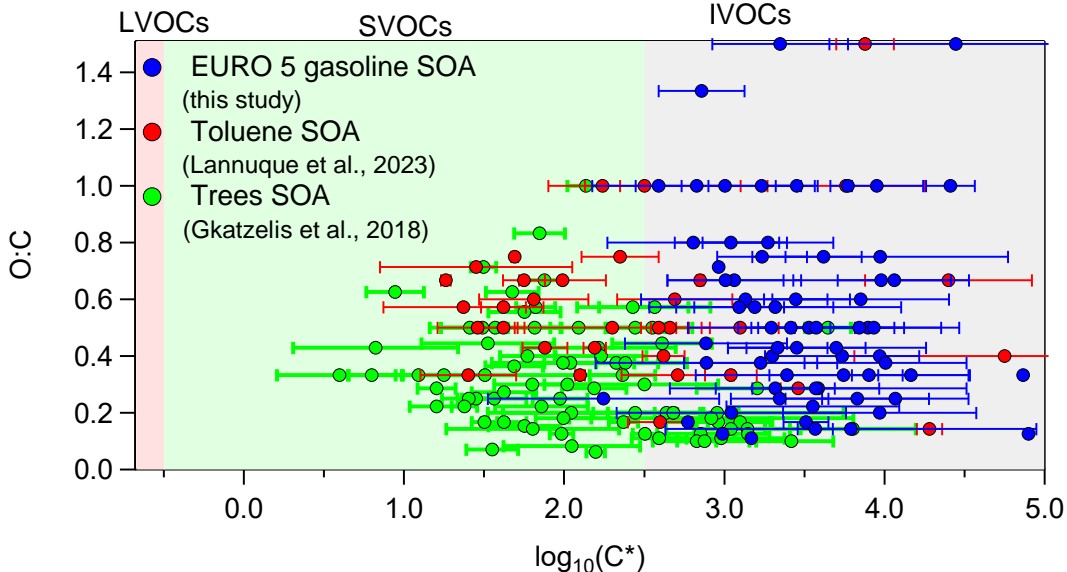

**Figure 6: Average volatility using experiment #3 (cold urban cycle) and experiment #5 (motorway cycle) compared to the average volatility found for toluene photo-oxidation SOA (Lannuque et al., 2023) and aged SOA derived from trees emissions ozonolysis (mixture of 42% δ3-carene, 38% α-pinene, 5% β-pinene, 4% myrcene, 3% terpinolene, and 8% other monoterpenes at 30 ±5 ºC)**
**(Gkatzelis et al., 2018). The error bars in the x axis correspond to the one standard deviation of the average experimental results.**






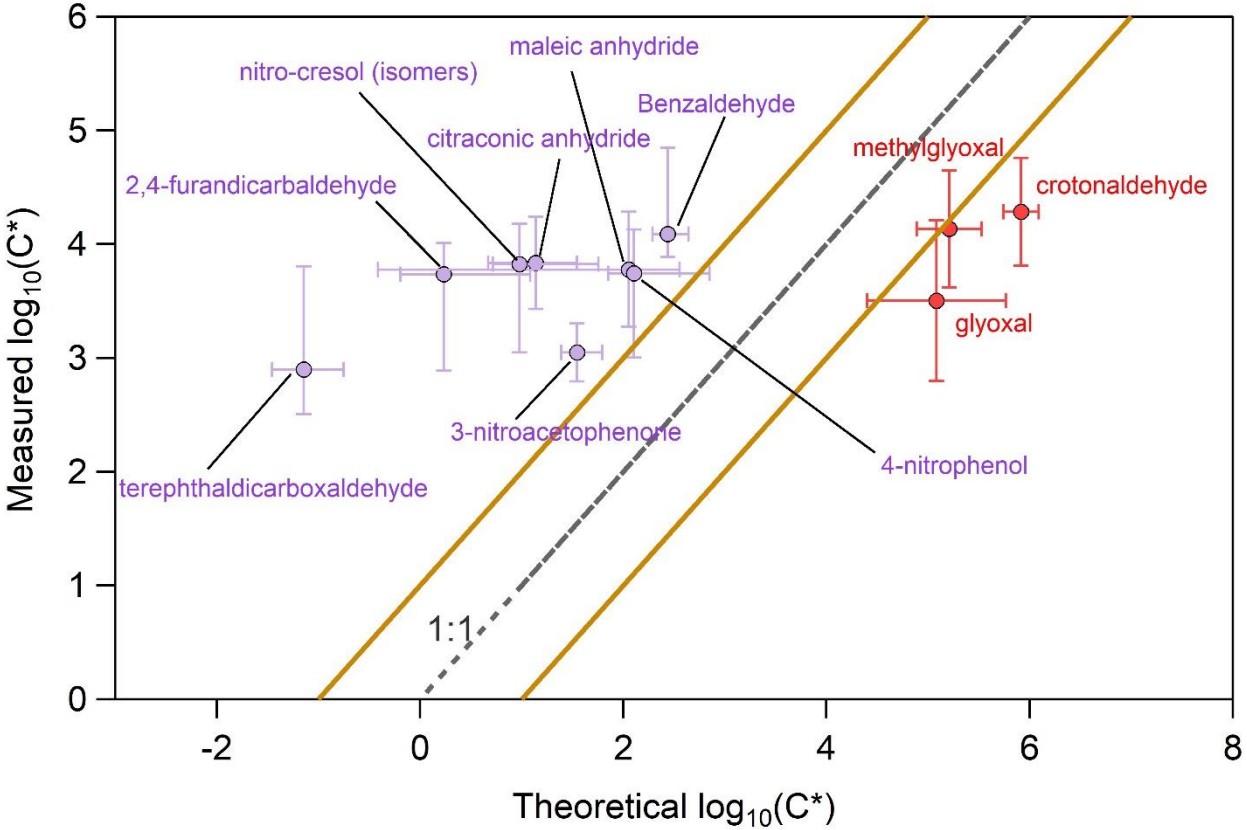

**Figure 7: Experimental and theoretical values of logC\* for 11 selected compounds. Several aldehydes are indicated with red cycles, while purple cycles correspond to all the other compounds. The error bars in the x axis correspond to the one standard deviation of the average theoretical values calculated from seven approaches, while the error bars in the y axis represent the one standard deviation of the average measured values. The grey dash line corresponds to the 1:1 line, while the brown lines denote a deviation of a logC\*±1 from the 1:1 line.**




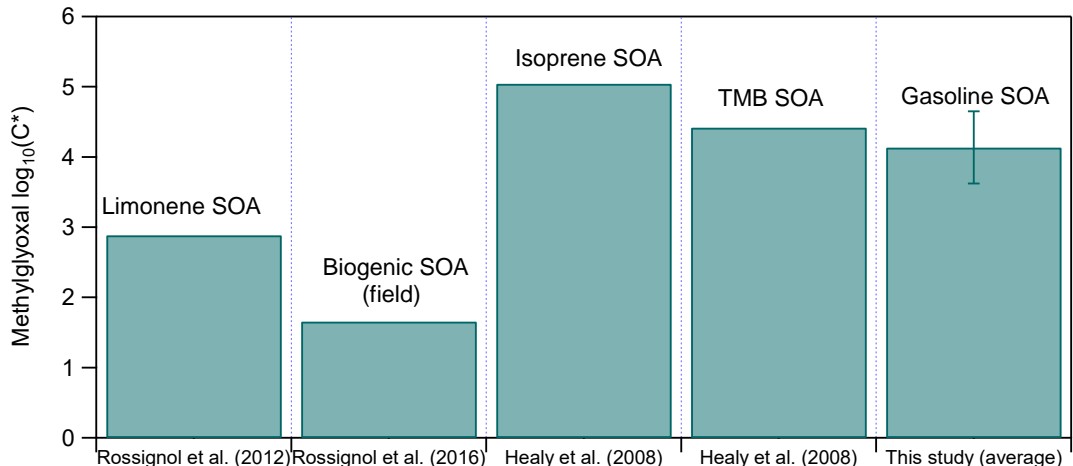

**Figure 8: Measured saturation concentration (in terms of logC\*) of methylglyoxal for different studies and systems.**

