# Peer review of "Secondary organic aerosol formed by EURO 5 gasoline vehicle emissions: chemical composition and gas-to-particle phase partitioning"

_EGUsphere, 2023_

## Author Comment (AC1)

'Secondary organic aerosol formed by EURO 5 gasoline vehicle emissions: chemical composition and gas-to-particle phase partitioning', Kostenidou et al.

Referee #1

This manuscript describes a study on the chemical composition of primary and photooxidation generated-secondary organic species (both gas and particle phases) from a EURO 5 gasoline under Artemis cold urban, hot urban, and motorway cycles. Gas and particle phase chemicals were analyzed by a chemical analysis of aerosol on-line (CHARON) inlet coupled with a proton-transfer-reaction time-of-flight mass spectrometer (PTR-ToF-MS). Gas-to-particle partitioning was presented as volatility distributions. I must admit that I am not an expert on organic aerosols. Therefore, I cannot provide expert opinion on the technical quality or novelty of this study. I do have some specific and technical comments as listed below.

Specific comments:

1. Section 2 Experimental. Page 4 Line 110-111: It will be worthwhile to explain the hypotheses for running the cold urban cycles at three different conditions (Experiments 1 to 3) as well as for using different dilution ratios. The result section then needs to explain if these hypotheses were tested true. Other than listing data in the tables or figures, the authors spent little effort to explain the differences from different cycles, either from the source of the differences or their real-world implication perspectives.

   For Exp 2 we inserted the whole cold urban cycle inside the chamber, while for Exp 1 and 3 we inserted only the first 5 minutes of the cycle. The reason for using only the first 5 minutes of the cycle is that most of the VOCs are emitted within this period according to Marques et al. (2022), who studied the fresh VOC emissions of this vehicle.

   Multiplying the fresh VOC emissions concentration with the corresponding partial dilution ratio (emissions to initial fresh VOCs into chamber, Table 1) we calculated the real fresh VOC concentrations to $338\times10^3$, $110\times10^3$ and $331\times10^3$ ppb for the experiments 1, 2 and 3 correspondingly. This result implies that inserting only the first 5 minutes of the cold urban cycle (experiments 1 and 3) the final VOC concentration is higher (approximately 3 times higher) compared to Exp 2, where the whole cold urban cycle was inserted. This is because if we continue inserting more emissions until the end of the cycle (15.3 minutes), the content of the chamber will be diluted, since this part of the emissions is characterized by very low VOCs levels and thus, the injection of a whole cycle will result in lower final VOCs concentrations inside the chamber. Indeed, the VOCs concentrations were lower for Exp 2, where the whole cold urban cycle was injected. However, this was not a hypothesis since we were aware of this from Marques et al. (2022).

   Concerning Exp 1 we did not apply any dilution after the emissions entered the chamber (i.e., DR was 1). Due to the high VOC concentrations inside the chamber, the signal of the major compounds at PTR-ToF-MS was saturated, thus we applied a dilution in front of the

PTR-ToF-MS inlet. This experiment indicated that the cold urban emissions in an 8 m$^3$ chamber from this vehicle were quite high to be measured by the PTR-ToF-MS. So, we decided that a dilution was needed for the rest experiments. Thus, for Exp 2 and 3 we applied dilutions of 3.2 and 5.1 respectively inside the chamber (using a pump and inserting purified air at the same moment, with the same volumetric rate) until the PTR-ToF-MS signal of the major VOCs was not anymore saturated. Thus, the different dilution ratios (after the emissions had entered the chamber) were a result of our effort to measure the fresh VOCs without signal saturation in the PTR-ToF-MS. The partial dilution ratio (emissions to initial fresh VOCs into chamber) was calculated by comparing the major emitted VOCs between the online fresh emissions (Marques et al., 2022) and the chamber emissions just after their injection into the chamber (we updated the values in Table 1, they were copied pasted from an old version accidentally).

The O and C distributions (VOCs, SVOCs and SOA) among the three different cycles were generally similar despite the small differences discussed already in the paper. What is significantly different among the cycles, is their emission factor levels. As it has been discussed in Marques et al., (2022) the cold urban emissions levels are approximately one order of magnitude higher than the corresponding motorway emissions. This suggest that the corresponding cold urban SOA will be significantly higher than the rest cycles. To verify this assumption, we calculated the production factors (PF) of the formed SOA (in µg km$^{-1}$) during each cycle using the equation bellow:

$$PF = SOA \; \frac{DR*V}{D}$$

where SOA is the mass concentration of the SOA, DR is the total dilution ratio after the vehicle exhaust, V is the volume of the chamber (8 m$^3$) and D is the distance of the injected cycle (4.51 km for a complete Artemis urban cycle, 1.53 km for the first 5 minutes of an Artemis urban cycle and 23.8 km for an Artemis motorway cycle). It must be mentioned that the above calculations do not include any wall-losses corrections. The production factors have been added to Table 1. A whole cold urban cycle produced 2096 µg km$^{-1}$ while a hot urban cycle resulted in almost the half concentration (982 µg km$^{-1}$) and a motorway cycle produced an SOA concentration 10 times lower (193 µg km$^{-1}$). The above demonstrates that the produced SOA in the cities, especially in the morning and in the afternoon, when people use their vehicles starting with a cold engine to drive to their workplaces and back, can be ten times higher compared to those produced by driving to a highway. Thus, the rural or suburban areas located near the highways are less affected by the SOA derived by gasoline GDI vehicles. In the revised version we added a small section: 4.4 SOA Production Factors and a sentence in the conclusions where we discuss all the above.

2. Table 1: Please describe why the VOC and NOx concentrations for Exp 1-3 are much higher than Exp. 4-5.

The VOC and NO$_x$ concentrations during the cold urban cycles (Exp 1-3) are much higher than those during the hot urban cycle (Exp 4) and motorway cycle (Exp 5) because of the lower catalyst efficiency during the cold urban cycle. Specifically, the first minutes of the

cold urban cycle the engine is still cold and thus the TWC efficiency is low. Online primary emissions measurements of the same vehicle obtained by Kostenidou et al. (2021) and Marques et al. (2022) confirm this behavior for PM, VOCs, $NO_x$ and Total Hydrocarbon (THC) concentrations. We added a sentence in the revised paper at the first paragraph of the Results section, explaining this behavior.

3. Page 5 Line 150. What are the sources for such high concentrations of ammonium nitrate? Section 4.3 mentioned that $NH_4NO_3$ particles may grew to >600 nm to clog the AMS orifice. It would be good to include size distribution evolution plots in supplemental materials, as the particle lifetime in the chamber as well as transmission efficiencies through inlets would change with particle size. Fig. 4 only shows two instant distributions, not time series.

High concentrations of ammonium nitrate were observed due to the high $NH_3$ and $NO_x$ amounts emitted by the tested vehicle especially during the cold urban cycles. $NO_x$ is oxidized to $HNO_3$ which reacts with $NH_3$ to form $NH_4NO_3$. Even though we only have $NO_x$ measurements (not $NH_3$) during this study, we do know that high $NO_x$ and $NH_3$ emissions is a general characteristic of the GDI vehicles as it is already stated in the manuscript (Page 9, lines 272-273).

However, we now added a clarification on page 5, line 150 and a second clarification on page 9, lines 272-273.

In addition, we added Figure S6 (Supplement) which illustrates the evolution of organics and nitrate mass distributions over Exp 2, where is clearly shown that the formed particles grow larger than 600 nm.

Technical corrections:

1. Abstract Page 1 Line 24, I would revise the sentence to: "Comparing our results to the theoretical estimations **for saturation concentrations**, we observed…"

Done.

2. Page 2 Line 57: Delete "…**the** those…"

Done.

3. Page 3 Line 67: Change "Except for" to "Besides"

Done.

4. Page 3 Line 68: Change "weather" to "whether"

Done.

5. Page 5, Line 129: missing a ")"

Done.

6. Page 5, Line 133: explain acronym "E/N"

E/N is the ratio of the electric field strength to the gas number density. It is now explained in the revised version.

7. 1: should TS include inorganic aerosol?

Yes, TS is the mass concentration of the total suspended particles and should include both organic and inorganic aerosol. Physically the total aerosol mass concentration reflects to the surface available for the semi-volatile organic compounds to condense on. We cannot ignore the inorganic particles (either externally or internally mixed with the organic particles) because they also offer a surface to the semi-volatile organics. The higher the total surface, the more organic material will be transferred to the particulate phase.

8. 2: explain the parameter $K_{p,i}$

The parameter $K_{p,i}$ is the gas-to-particle partitioning coefficient. It is now explained in the revised manuscript.

9. Page 14, Line 431: missing a "("

We added a "(".

10. 1-3: the y-axis's are fractions rather than %.

We now change the y-axis names of Figures 1-3 from "%" to "Fraction of".

References:

Kostenidou, E., Martinez-Valiente, A., R'Mili, B., Marques, B., Temime-Roussel, B., Durand, A., André, M., Liu, Y., Louis, C., Vansevenant, B., Ferry, D., Laffon, C., Parent, P., and D'Anna, B.: Technical note: Emission factors, chemical composition, and morphology of particles emitted from Euro 5 diesel and gasoline light-duty vehicles during transient cycles, Atmos. Chem. Phys., 21, 4779–4796, 2021.

Marques, B., Kostenidou, E., Martinez-Valiente, A., Vansevenant, B., Sarica, T., Fine L., Temime-Roussel, B., Tassel, P., Perret, P., Liu, Y., Sartelet, K., Ferronato, C., and D'Anna B.: Detailed speciation of non-methane volatile organic compounds in exhaust emissions from diesel and gasoline Euro 5 vehicles using online and offline measurements, Toxics, 10, 4, 184, https://doi.org/10.3390/toxics10040184, 2022.

---

## Author Comment (AC2)

'Secondary organic aerosol formed by EURO 5 gasoline vehicle emissions: chemical composition and gas-to-particle phase partitioning', Kostenidou et al.

Referee #2

The authors applied for the first time PTRMS equipped with a CHARON device to the photooxidation products of gasoline vehicle exhaust and measured the molecular distribution of precursor gases, product gases, and product particles. The detected gas and particle products were consistent with the results of previous chemical analyses of aromatic hydrocarbon chamber experiments. The saturation concentrations of selected products were estimated from the gas-to-particle ratios. The average saturation concentration of secondary organic aerosol (SOA) particles was evaluated to be higher for lower organic aerosol concentrations, consistent with the gas/particle partitioning model. The saturation concentrations evaluated by present experiments were compared with theoretical predictions. A novel aspect of this study is the quantitative evaluation of gas particle distribution at the molecular level with respect to the subject gasoline vehicle SOA. However, the current manuscript does not adequately discuss the uncertainties in the sensitivity ratios of gas and particle analytes that may interfere with the evaluation results. There could be a more in-depth discussion of fragmentation as well. Therefore, this manuscript can be expected to be published but needs to be revised.

(1) Line 24 (abstract). What is "theoretical estimations"? Could you add explanations?

We modified the sentence in: "… the theoretical estimations for saturation concentrations…"

(2) Lines 48-50. Morino et al. (2022) should be added as a recent paper that experimentally discussed photooxidation products in gasoline vehicle exhaust.

Ref. Morino, Y., Li, Y., Fujitani, Y., Sato, K., Inomata, S., Tanabe, K., Jathar, S.H., Kondo, Y., Nakayama, T., Fushimi, A., Takami, A., Kobayashi, S. Secondary organic aerosol formation from gasoline and diesel vehicle exhaust under light and dark conditions, Environ. Sci.: Atmos., 2, 46-64, 2022.

We have added the proposed reference to the revised manuscript.

(3) Lines 163-164 and 417-428: Does the statement in lines 163-164 mean that there is a maximum uncertainty of about 100 in the ratio of the CHARON-PTRMS signal to the PTRMS signal? Is it correct that there is an error of ±2 in the logC* measured in this case? If the experimental error is ±2, many experimental results agree with theory within the error range, and discussion described at lines 417-428 might be too much detailed. Other than the cited reference, is there any experimental evidence that would allow a specific discussion regarding the uncertainty of the ratio of the CHARON-PTRMS signal to the PTRMS signal for present measurement subject compounds?

The sentence in Lines 163-164: "*The OA mass concentrations of HR-ToF-AMS and CHARON may differ between each other up to a factor of 2 as mentioned in Müller et al. (2017) due to fragmentation of analyte ions in PTR-ToF-MS*" means that the mass concentration measured by CHARON could be up to two times lower than the mass concentration measured by AMS. For example, if HR-ToF-AMS measures 3 µg m$^{-3}$ then CHARON measures 1.5 µg m$^{-3}$ (not 0.03 µg m$^{-3}$). In our study the organic HR-ToF-AMS mass concentration was 1.1-1.5 times higher than the organic CHARON mass concentration. This difference could be due to the fragmentation in PTR-ToF-MS but also because of the different cut of size at the low size range for each instrument (100-150nm for CHARON and 50-70nm for HR-ToF-AMS). For this reason, we scaled up organic CHARON concentration based on organic AMS mass concentration, to "correct" CHARON concentration. This is stated in a sentence just before: "…*while the corresponding particle phase (Cp,i) concentration was measured by CHARON, after normalizing the total OA CHARON mass concentration to the total HR-ToF-AMS OA mass concentration.*" However, we modified these sentences to be better understandable.

To our knowledge a direct mass concentration comparison between HR-ToF-AMS and CHARON for specific compounds and any linkage to fragmentation has not been performed. The fragmentation in PTR-ToF-MS differs for each compound and is more intense as E/N increases (e.g., Müller et al., 2017; Leglise et al., 2019). Leglise et al. (2019) showed that the fragmentation of certain compounds at E/N 100 Td could be that high, that zero signal is left at the parent m/z. However, most of the tested compounds in Leglise et al. (2019) are not related to our study. Using standard compounds expected to be produced in our experiments, we examined their fragmentation in both CHARON and PTR-ToF-MS modes, and we concluded that the signal at the parent m/z may vary between 12 and 100 % of the total signal. In the table below we present the % of the total signal at the parent m/z compounds expected to be found in formed gasoline SOA (at E/N =100 Td):

| Compound | Parent m/z | % of the total signal PTR-ToF-MS | % of the total signal CHARON |
|---|---|---|---|
| Maleic anhydride | 99.01 | 95.9 | 91 |
| Hydroquinone | 109.03 | 88.6 | 87.2 |
| 5-Methylfurfural | 111.04 | 88.4 | 86.2 |
| Benzoquinone | 109.03 | 88.6 | 100 |
| Nonanal | 143.3 | 50.8 | 51.9 |
| Heptanal | 115.2 | 15.7 | 21.8 |
| Hexanal | 101.2 | 11.9 | 15.6 |
| Pentanal | 87.1 | 12.2 | 16 |

The $C_{g,i}$ concentration is linked to the PTR-ToF-MS measurement uncertainty, which depends on the compound and could be up to 30-40%. The $C_{p,i}$ concentration depends on all possible uncertainties of PTR-ToF-MS detection as the compound was in the gas phase and in addition to any uncertainties related to transmission in the aerodynamic lens of CHARON and the error in vaporization (up to 50% depending on the compound and the particle size). We assume that the fragmentation in PRT-ToF-MS and CHARON (i.e., CHARON + PTR-ToF-MS) for a compound does not differ substantially. TS (total aerosol mass concentration) is provided by the HR-ToF-

AMS and so its uncertainty it's the HR-ToF-AMS measurement uncertainty (up to 30-40%). Thus, the logC$^*$ uncertainty is a function of the two instruments measurement uncertainties.

So, the error in the logC* is much lower than ±2 and so, the comparison with the theoretical values and logC$^*$ values from other studies can be safely discussed. We have added some discussion about the uncertainty of the logC$^*$ in the revised manuscript (section 4.3).

(4) Lines 311-324. With respect to nitroaromatics, there may be underestimation due to fragmentation compared to aldehydes and ketones. If available, please discuss any experimental information on fragmentation of nitroaromatics. The authors assume that heterogeneous processes are important for the formation of nitroaromatic hydrocarbons, but there is also a hypothesis that nitrophenols are formed by gas-phase reactions of phenoxy-type radicals with $NO_2$ (e.g. Harrison et al., 2005). Is there any evidence from the results of this study to support the hypothesis of heterogeneous reactions?

Ref. Harrison, M.A.J., Barrra, S., Borghesi, D., Voine, D., Arsene, C., Olariu, R.I., Nitrated pheols in the atmosphere: a review, Atmos. Environ., 39, 231-248, 2005.

We examined the fragmentation of some pure compounds in CHARON mode. Among them, there were some aldehydes and some nitroaromatics. Specifically, we found that:

for methylglyoxal 86.6% of the total signal was attributed to the parent m/z 73.03,

for 5-methylfurfural 86.2% of the total signal was attributed to the parent m/z 111.04,

for 4-nitrophenol 91.2% of the total signal was attributed to the parent m/z 140.03 and

for nitrocatechol 72.4% of the total signal was attributed to the parent m/z 156.03

4-nitrophenol and nitrocatechol did not fragment significantly and their signal at the parent ion (73-91%) was comparable to those of the functionalized aldehydes such as methylglyoxal (~87%) which had an appreciable contribution in the formed SOA. Thus, nitroaromatics were likely not underestimated compared to the functionalized aldehydes. Nevertheless, to be clear about fragmentation we have already stated in the paper that "Tables 4 and S5 do not account for any fragmentation in the concentration calculations unless it is mentioned." In the revised version we added the same statement for the SVOC (Tables 3 and S3).

From the data of this study, we cannot support whether the nitroaromatics were formed through heterogeneous or gas-phase reactions. Their gas and particle phase concentrations do not differ importantly, thus there are no evidence to support one way or the other.

(5) Lines 382-384. The author discusses the cause of the experimental results, but it is not clear. Do the authors want to write that the experimental results are qualitatively explained by the gas/particle partitioning model?

Yes indeed. According to the gas/particle partitioning model, when the formed aerosol has relatively lower concentration most of the semi-volatile compounds will not reach the corresponding saturation concentration level and thus, they will remain in the gas phase. Increasing the concentration of the formed OA the atmosphere gradually becomes saturated and when the aerosol concentration of an x compound exceeds its saturation concentration this compound will start partitioning between the gas and the particle phase. This behavior characterizes all the semi-volatile compounds that constitute SOA, but each compound has different saturation concentration.

At the same time the existence of pre-existing particles facilitates the condensation of the vapors because particles serve as surface for the vapors to condense on. This is the reason of the ammonium sulfate particles are being used in environmental chamber experiments. In the absence of pre-existing particles, the first semi-volatile products will condense onto the chamber walls, and they will be never measured. Since, there are always pre-existing particles in the ambient, laboratory simulations should include also pre-existing particles (usually ammonium sulfate).

Thus, one reason for the lower volatility of the SOA that Lannuque et al. (2022) reported is probably due to the lower aerosol concentration (toluene SOA and pre-existing ammonium sulfate), which allows a larger fraction of the higher volatility compounds to remain in the gas phase. When the SOA mass concentration increases (as in our work) a larger fraction of the higher volatility compounds is forced to move to the particle phase and so SOA contains a larger part of higher volatility compounds. We now added a sentence clarifying this behavior.

(6) Line 431. There is an end of parentheses, but the beginning of parentheses is unclear.

We added a "(".

References:

Leglise, J., Müller, M., Piel, F., Otto, T. and Wisthaler, A.: Bulk organic aerosol analysis by proton-transfer-reaction mass spectrometry: an improved methodology for the determination of total organic mass, O:C and H:C elemental ratios, and the average molecular formula, Anal. Chem., 91, 12619–12624, 2019.

Müller, M., Eichler, P., D'Anna, B., Tan, W., and Wisthaler, A.: Direct sampling and analysis of atmospheric particulate organic matter by Proton-Transfer-Reaction Mass Spectrometry, Anal. Chem., 89, 10889-10897, 2017.

Lannuque, V., D'Anna, B., Kostenidou, E., Couvidat, F., Martinez-Valiente, A., Eichler, P., Wisthaler, A., Müller, M., Temime-Roussel, B., Valorso, R., and Sartelet, K.: Gas-particle partitioning of toluene oxidation products: an experimental and modeling study, EGUsphere [preprint], https://doi.org/10.5194/egusphere-2023-1290, 2023.